# Estrogen exacerbates mammary involution through neutrophil-dependent and -independent mechanism

Chew Leng Lim[1,2†], Yu Zuan Or[2†], Zoe Ong[2], Hwa Hwa Chung[2], Hirohito Hayashi[3], Smeeta Shrestha[4], Shunsuke Chiba[3], Feng Lin[5], Valerie Chun Ling Lin[2]*

[1]NTU Institute for Health Technologies, Interdisciplinary Graduate School, Nanyang Technological University, Singapore, Singapore; [2]School of Biological Sciences, Nanyang Technological University, Singapore, Singapore; [3]Division of Chemistry and Biological Chemistry, School of Physical and Mathematical Sciences, Nanyang Technological University, Singapore, Singapore; [4]School of Basic and Applied Sciences, Dayananda Sagar University, Bangalore, India; [5]School of Computer Science and Engineering, Nanyang Technological University, Singapore, Singapore

*For correspondence:
cllin@ntu.edu.sg

†These authors contributed equally to this work

Competing interests: The authors declare that no competing interests exist.

**Abstract** There is strong evidence that the pro-inflammatory microenvironment during post-partum mammary involution promotes parity-associated breast cancer. Estrogen exposure during mammary involution drives tumor growth through neutrophils' activity. However, how estrogen and neutrophils influence mammary involution are unknown. Combined analysis of transcriptomic, protein, and immunohistochemical data in BALB/c mice showed that estrogen promotes involution by exacerbating inflammation, cell death and adipocytes repopulation. Remarkably, 88% of estrogen-regulated genes in mammary tissue were mediated through neutrophils, which were recruited through estrogen-induced CXCR2 signalling in an autocrine fashion. While neutrophils mediate estrogen-induced inflammation and adipocytes repopulation, estrogen-induced mammary cell death was via lysosome-mediated programmed cell death through upregulation of *cathepsin B*, *Tnf* and *Bid* in a neutrophil-independent manner. Notably, these multifaceted effects of estrogen are mostly mediated by ERα and unique to the phase of mammary involution. These findings are important for the development of intervention strategies for parity-associated breast cancer.

## Introduction

There is strong evidence that the mammary microenvironment during the post-partum mammary involution promotes mammary tumor progression. High levels of tissue fibrillar collagen and elevated expression of cyclooxygenase-2 (COX-2) in the mammary gland have been shown to drive tumor growth and lymph angiogenesis (*Lyons et al., 2014*; *Lyons et al., 2011*). Wound healing-like tissue environment associated with mammary involution is also known to promote tumor development and dissemination (*Foster et al., 2018*). Estrogen has been shown to stimulate the growth of estrogen receptor-negative mammary tumors during mammary involution and estrogen-stimulated neutrophil activity plays a crucial role in fostering the pro-tumoral microenvironment (*Chung et al., 2017*). This suggests that estrogen exposure during post-weaning mammary involution is a risk factor for parity-associated breast cancer. However, the functional roles of estrogen and neutrophils in mammary biology during involution have been little studied to date.

Post-weaning mammary involution is a process for the lactating mammary gland to return to the pre-pregnancy state. The distinctive features of mammary involution include massive cell death of the secretory mammary alveoli, acute inflammation, extracellular matrix remodelling and adipocyte repopulation. Involution is commonly divided into two phases. In mice, the first phase is the

reversible phase whereby the reintroduction of the pups within 48 hr can re-initiate lactation (*Li et al., 1997*; *Jaggi et al., 1996*). It is typified by a decrease in milk protein synthesis and increased mammary cell death resulting in the appearance of shed, dying cells within the lumen of the distended alveoli (*Strange et al., 1992*; *Walker et al., 1989*). Inflammation also occurs in the first phase with the infiltration of immune cells and up-regulation of immune response genes (*Stein et al., 2004*; *Clarkson and Watson, 2003*). The second, and irreversible phase of mammary involution occurs after 72 hr of weaning. This phase is morphologically characterized by the collapse of the alveolar structure, the second wave of epithelial cell death, continued inflammation and the repopulation of adipocytes. This is followed by mammary regeneration and tissue remodelling to the pre-pregnancy state.

Neutrophils are the most abundant leukocytes in the innate immune system against invading pathogens. Neutrophils are also activated in response to sterile inflammation, but the outcome is complicated and depends on the context (*Wang, 2018*). The importance of neutrophils in tumor development has been increasingly recognized in recent years. Tumor-associated neutrophils have been classified into the anti-tumor neutrophils (N1) and pro-tumor neutrophils (N2) based on the expression of specific markers (*Fridlender et al., 2009*). Neutrophils are known to accumulate in the peripheral blood of patients with cancer and a high circulating neutrophil-to-lymphocyte ratio is known as a strong biomarker of poor prognosis in various cancers (*Shaul and Fridlender, 2019*). Based on density gradient centrifugation, circulating neutrophils in mice model can also be classified into the low-density and high-density neutrophils (LDN and HDN) (*Sagiv et al., 2015*; *Mishalian et al., 2017*). LDN was associated with immunosuppressive activity promoting tumor growth while HDN was considered cytotoxic to the tumor. Neutrophils were reported to be the first immune cells recruited into mammary tissue during involution (*Stein et al., 2004*), although the significance of their presence is not known.

Estrogen is well studied on its mitogenic effect and plays essential roles in mammary ductal development. However, the studies of estrogen influence on mammary involution are scarce. In mice, a histological examination in 1970 reported that estrogen treatment retards mammary involution (*Satoh, 1970*). However, estrogen was reported to promote the regression of mammary glands in rats by up-regulating a 60K gelatinase which degrades collagen and participate in extracellular matrix remodelling (*Ambili et al., 1998*). Similar to the observation in rats, injection of estrogen during the dry off period in dairy cows hastened involution based on the abrupt decline of milk synthesis and secretion (*Athie et al., 1997*; *Athie et al., 1996*). Those early preliminary studies clearly indicate the involvement of estrogen in the modulation of mammary involution, but the pieces of evidence are limited and inconsistent among different models.

In this present study, we investigated the influence of estrogen on the progression of mammary involution in mice. We also evaluated the roles of neutrophils in estrogen regulation of this process in neutrophil depletion experiments. Our data show that estrogen plays a multifaceted role in the regulation of post-weaning mammary involution by enhancing inflammation, cell death, adipocyte occupancy, and tissue remodelling through both neutrophil-dependent and neutrophil-independent mechanisms. Global transcriptomic analysis revealed striking effect of estrogen on neutrophil activities that likely exert profound influence on mammary microenvironment that may have significantly impact on the development of parity-associated breast cancer.

## Results

### Estrogen promotes inflammation, programmed cell death and adipocytes repopulation during post-weaning mammary involution

Acute inflammation, programmed cell death, and adipocytes repopulation are the hallmarks of the acute phase of post-weaning mammary involution. To understand the roles of estrogen in mammary involution, its effect on these hallmarks were evaluated. Ovariectomized (OVX) mice at 24 hr post-weaning (INV D1) were treated with or without 17β-estradiol benzoate (E2B) for 48 hr. E2B caused significantly more mammary cell death, as evidenced by the increase of the number of shed cells with hyper-condensed nuclei in the lumen, a characteristic of programmed cell death (*Figure 1*, Ai, and 1B, p=0.0131). This effect was associated with an increase in the number of cleaved caspase-3-positive (CC3+) mammary cells (*Figure 1*, Aii, and 1B, p=0.0283). E2B also significantly increased the

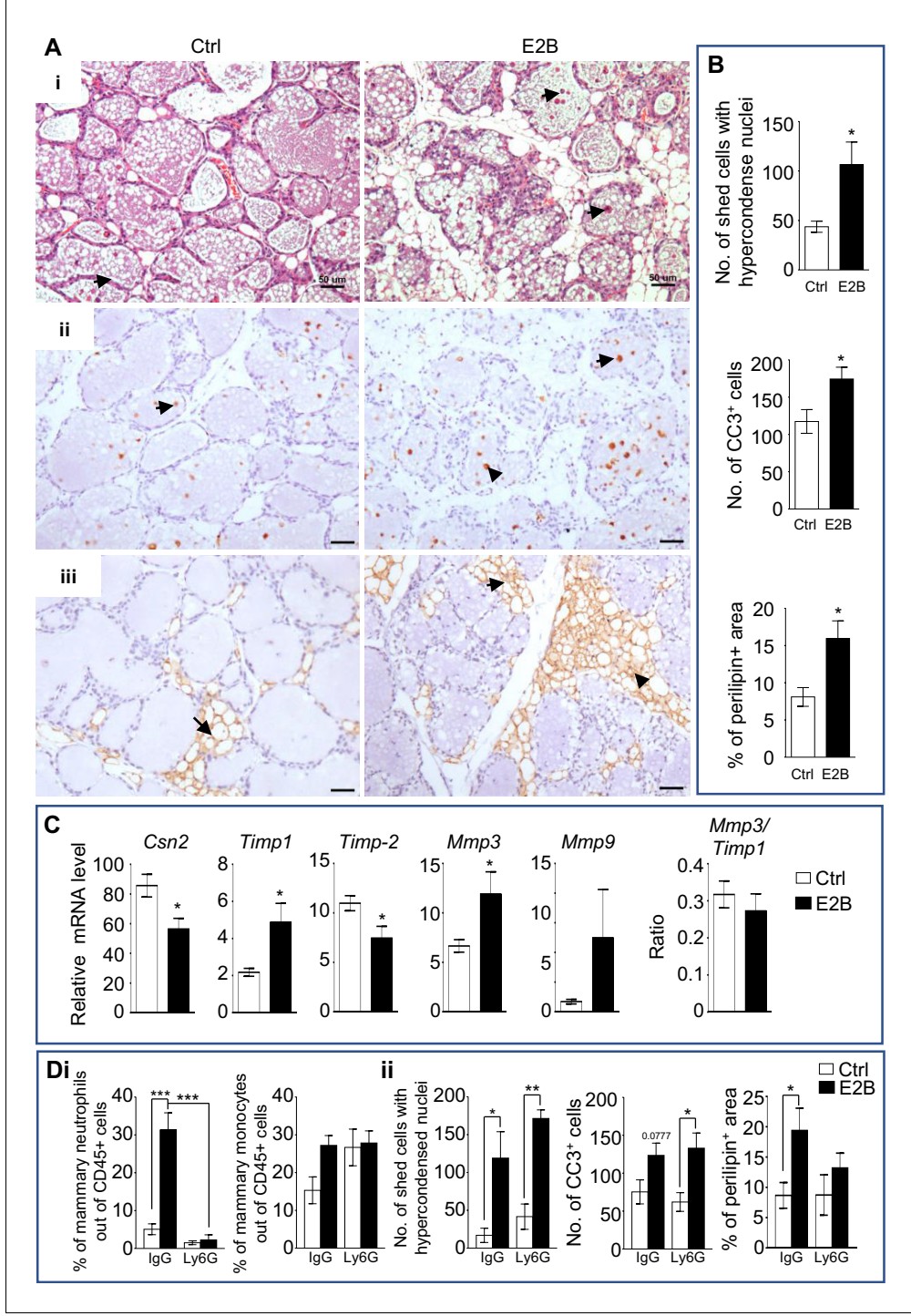

**Figure 1.** Estrogen accelerates mammary involution. (A-C) Mice on the day of weaning (involution day 1-INV D1) were treated with vehicle control (Ctrl) or E2B for 48 hr before mammary tissues were collected for analysis; (A) (i) H and E stained mammary tissue sections; shed cells with hyper-condensed nuclei are indicated by arrows; (ii) IHC of cleaved caspase-3 (CC3); arrows indicate CC3+ cells; (iii) Perilipin IHC; arrows indicate perilipin+ adipocytes; Scale bars: 50 μm; (B) The number of shed cells with hyper-condensed nuclei in the lumens of H and E sections is based on the total of 5 random field per mouse (Ctrl n = 9, E2B n = 8); the number of CC3+ cells is based on the average of CC3+ cells of 5 random field per mouse (Ctrl n = 7, E2B n = 6); adipocytes repopulation is based on the percentage of perilipin stained area of the total section (Ctrl n = 7, E2B n = 8); (C) Gene expression of *Csn2* and tissue remodelling enzymes *Timp1*, *Timp2*, *Mmp3*, and *Mmp9* relative to *Rplp0* by qPCR analysis (Ctrl n = 7,

*Figure 1 continued on next page*

*Figure 1 continued*

E2B n = 6). (D) Mice at INV D1 was treated with anti-Ly6G antibody (Ly6G) or isotype control (IgG). 6 hr later, they were treated with vehicle control (Ctrl) or E2B for 48 hr; Di, E2B treatment in mice given IgG significantly increased the percentage of mammary neutrophils by 9-folds which was abolished by neutrophil depletion with Ly6G; Percentage of mammary neutrophils (CD45+ CD11b+ Gr1+ F4/80-) and monocytes (CD45+ CD11b+ Ly6C$^{hi}$) out of live CD45+ population; Dii, Neutrophil depletion had no effect on cell shedding and number of Cc3$^+$ cells, but attenuated estrogen-induced adipocytes repopulation (Ctrl+IgG n = 4, E2B+IgG n = 4, Ctrl+Ly6G n = 3, E2B +Ly6G, n = 3). Data represented as mean ± SEM.

The online version of this article includes the following figure supplement(s) for figure 1:

**Figure supplement 1.** Effect of neutrophil depletion on estrogen-induced cell death and adipocytes repopulation.

repopulation of adipocytes based on the staining of perilipin, a lipid droplet-associated protein (*Greenberg et al., 1991*; *Figure 1*, Aiii, and 1B, p=0.0155). Accordingly, E2B decreased the expression of milk proteins β-casein (*Csn2*) (*Figure 1C*, p=0.0187).

Coordinated activities of metalloproteinases and the tissue inhibitor of metalloproteinases are key players in mammary tissue remodelling and adipocyte repopulation during mammary involution. In particular, stromelysin-1 (*Mmp3*) gene deletion or *Timp1* overexpression in mice accelerated mammary adipogenesis (*Barker et al., 2011*; *Alexander et al., 2001*). Intriguingly, E2B-induced adipocyte repopulation was associated with increases in the gene expression of *Timp1*, *Mmp3* and *Mmp9*, but decrease in the expression of *Timp2*. However, the ratio of *Mmp3* to *Timp1* did not change in response to E2B (*Figure 1C*). It is plausible that calibrated activities of MMPs and their inhibitors are involved in E2B-induced adipocyte repopulation and tissue remodeling.

We reported previously that E2B markedly induced the expression of inflammatory genes and neutrophil infiltration (*Chung et al., 2017*). We questioned if neutrophils are involved in E2B-induced cell death and adipocytes repopulation. The effect of estrogen on mammary involution following neutrophil depletion using anti-Ly6G antibody (Ly6G) were evaluated. *Figure 1Di* shows that estrogen increased mammary neutrophils significantly (p=0.0002). Ly6G antibody reduced neutrophils in the mammary tissue of E2B-treated samples by ~90%. Estrogen also visibly increases mammary monocytes but not to a statistically significant level (*Figure 1Di*, p=0.1103). Interestingly, whilst neutrophil depletion had no effect on E2B-induced mammary cell death, it attenuated E2B-induced adipocytes repopulation (*Figure 1Dii*). The representative histological images of the effect of neutrophil depletion on cell death and adipocytes repopulation are shown in *Figure 1—figure supplement 1*. Taken together, we conclude that estrogen promotes mammary involution, and neutrophils are critical for estrogen-induced inflammation and adipocytes repopulation, but not for mammary cell death.

## Majority of estrogen-regulated genes in involuting mammary tissue is mediated through neutrophils

To elucidate the mechanism of estrogen regulation during mammary involution, RNA-Seq analysis of mammary tissue was conducted. Since estrogen has been shown to induce neutrophil infiltration in mammary tissue, the involvement of neutrophils in estrogen regulation of gene expression was also determined. OVX mice were injected with anti-Ly6G antibody (Ly6G) to deplete neutrophils at 24 hr post-weaning (INV D1). Mice given isotype control antibody (IgG) were used as controls. Mice were subsequently treated with or without E2B for 24 hr. Consistent with the previous study (*Chung et al., 2017*), E2B induced the infiltration of neutrophil and monocytes (CD45+ CD11b+ Ly6C$^{hi}$) in mammary tissue during mammary involution. (*Figure 2—figure supplement 1B*). Anti-Ly6G treatment reduced neutrophil (CD45+ CD11b+ Gr1$^{hi}$) levels in the blood and mammary tissue by more than 95% (*Figure 2—figure supplement 1A*, p<0.05). Mammary monocytes were decreased by anti-Ly6G, although this reduction was not statistically significant (*Figure 2—figure supplement 1B*, p=0.1417). It is plausible that E2B-induced monocytes infiltration is partly mediated by neutrophils.

RNA-Seq data were analyzed using DESeq2 (*Love et al., 2014*) to identify differentially expressed (DE) genes between the different treatments. As shown in the volcano plot (*Figure 2A*, top panel), a total of 1987 genes were significantly (padj < 0.05) regulated by E2B in IgG groups with very high

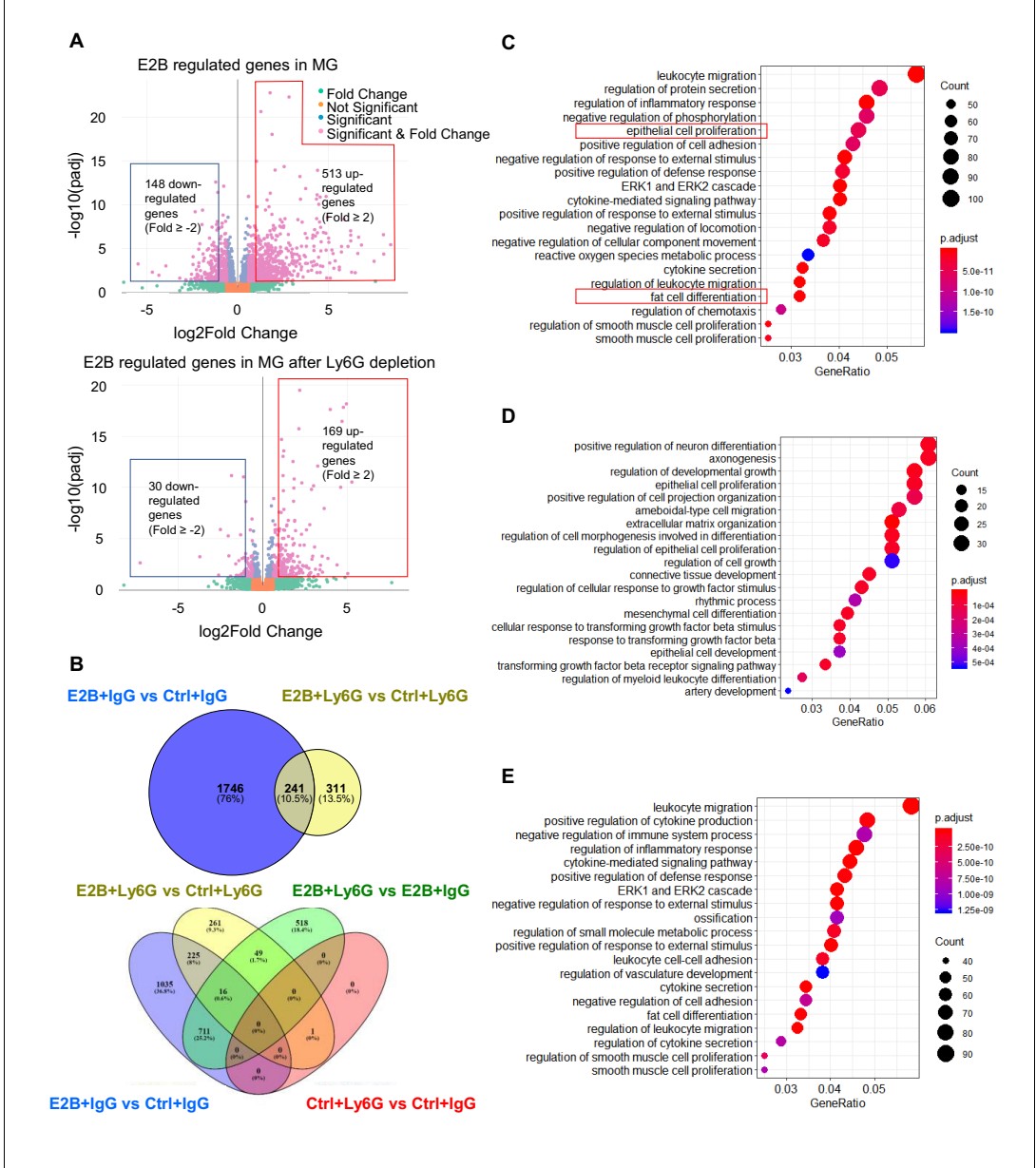

**Figure 2.** Estrogen regulates a multitude of neutrophil-dependent and –independent biological processes in involuting mammary gland. Mice at INV D1 were treated with anti-Ly6G antibody (Ly6G) or isotype control (IgG). 24h later, they were treated with vehicle control (Ctrl) or E2B for 24h (Ctrl+IgG n=3, Ctrl+Ly6G n=3, E2B+IgG n=3, E2B+Ly6G n=3). RNA-Seq data were processed and analyzed with DESeq2 followed by GO over-representation analysis. (A) Volcano plot for the differentially expressed E2B regulated genes in mammary gland (MG) from IgG- and Ly6G-treated animals. (B) Venn diagram for the differentially expressed genes identified from the DESeq2 analysis of the RNA-Seq data. (C) Top 20 Gene Ontology (GO) terms for E2B regulated genes in MG without neutrophil depletion. (D) Top 20 GO terms for the E2B regulated genes in MG after neutrophil depletion. E, Top 20 GO terms for E2B regulated genes lost as a result of neutrophil depletion.

The online version of this article includes the following source data and figure supplement(s) for figure 2:

**Source data 1.** DESeq2 analysis result between Ctrl+IgG and Ctrl+Ly6G.
**Source data 2.** DESeq2 analysis result between E2B+IgG and Ctrl+IgG.
**Source data 3.** DESeq2 analysis result between E2B+Ly6G and Ctrl+Ly6G.
**Source data 4.** DESeq2 analysis result between E2B+IgG and E2B+Ly6G.
**Source data 5.** GO over-representation analysis result of DE genes between E2B+IgG and Ctrl+IgG.
**Source data 6.** GO over-representation analysis result of DE genes between E2B+Ly6G and Ctrl+Ly6G.
**Source data 7.** GO over-representation analysis result of DE genes found only in E2B+IgG.
**Figure supplement 1.** Depletion efficiency of neutrophils with anti-neutrophil antibody Ly6G.

fold changes (*Figure 2B*). Of these genes, 513 genes were up-regulated, and 148 genes were down-regulated with a fold change $\geq 2$ (*Figure 2A*, top panel). Gene ontology (GO) analysis showed that estrogen regulates genes involved in diverse biological processes (*Figure 2C*). Leukocyte migration, regulation of inflammatory response, and cytokine-mediated signalling are among the top biological processes regulated. E2B also regulated genes in epithelial cell proliferation, cell death, and fat cell differentiation.

Remarkably, neutrophil depletion eliminated 88% of estrogen-regulated genes (*Figure 2B*, top panel), that is, 1746 out of 1987 E2B-regulated genes in mammary tissue were regulated in neutrophils directly, or indirectly through the activities of neutrophils. Correspondingly, 12% (241 genes) of E2B-regulated genes in mammary tissue are independent of neutrophils. It is also noteworthy that there is only one differentially regulated gene (padj < 0.05) between IgG and Ly6G groups in the absence of estrogen (*Figure 2B*, bottom panel). This could be due to the fact that the number of mammary neutrophils present in these Ctrl samples were too low (less than 1% of the total live cells in FACS analysis) to account for any differences in gene expression. Consistent with the function of neutrophils, E2B-regulated GO terms including leukocyte migration, inflammatory response, cytokines and chemokines related pathways, and fat cell differentiation were all eliminated in neutrophil depleted samples (*Figure 2D*). On the other hand, GO terms such as epithelial cell proliferation and developmental growth persisted in neutrophil-depleted samples, suggesting that their stimulation by E2B are independent of neutrophils (*Figure 2C and D*).

Taken together, global gene expression analysis of mammary tissue demonstrates that estrogen regulates genes in a plethora of biological processes during post-weaning mammary involution, and 88% of estrogen-regulated genes are mediated in neutrophils, or through neutrophils' influence on other cells such as macrophages or monocytes. Therefore, the E2B-induced pro-inflammatory micro-environment during mammary involution is primarily due to its regulation of neutrophil activities, which play a significant role in post-weaning mammary involution. In the subsequent studies, we investigated the mechanism of estrogen-induced neutrophil infiltration, mammary cell death, and adipocytes repopulation during post-weaning mammary involution.

## Estrogen-induced *Cxcr2* signalling in neutrophils plays a key role in mammary neutrophil infiltration

Based on the GO over-representation analysis, 63 E2B-regulated genes with fold change $\geq 3$ were associated with leukocyte migration and inflammation (*Figure 3A*). All 63 genes were no longer E2B-regulated after neutrophil depletion (in Ly6G group). This suggests that the regulation of inflammation by estrogen is primarily exerted through neutrophils. Aconitate decarboxylase 1 (*Acod1*), triggering receptor expressed on myeloid cells 1 (*Trem1*), triggering receptor expressed on myeloid cells 3 (*Trem3*), and interleukin one beta (*Il1b*) are among the top up-regulated genes based on fold induction. These genes were validated with qPCR in isolated neutrophils (*Figure 3B*, *Acod1*, p=0.0845; *Trem1*, p=0.0252; *Trem3*, p=0.0181; *Il1b*, p=0.0527). Another interesting estrogen-regulated gene is myocardial infraction associated transcript 2 (*Mirt2*; *Figure 3B*, p=0.008), which was found to be one of the highest E2B-regulated gene in the RNA-Seq data. *Mirt2* is a lipopolysaccharides (LPS)-induced long non-coding RNA (lncRNA) in macrophages and was reported to be a negative regulator of LPS-induced inflammation both in vivo and in vitro (*Du et al., 2017*). Thus, *Mirt2* upregulation by E2B in neutrophils indicates its involvement in the moderation of estrogen-induced inflammatory response during mammary involution.

Consistent with a previous report (*Chung et al., 2017*), S100 calcium-binding protein A8 (*S100a8*) and A9 (*S100a9*) were all induced by E2B in neutrophils and validated by qPCR (*Figure 3A and B*). More interestingly, C-X-C motif chemokine receptor 2 (*Cxcr2*) and its known ligands in the mouse (*Bachelerie et al., 2019*), C-X-C motif chemokine ligand *Cxcl1, Cxcl2, Cxcl3* and *Cxcl5* were all upregulated by E2B (*Figure 3A*). Furthermore, qPCR analysis verified that estrogen upregulated the expression of *Cxcr2, Cxcl2, Cxcl3* and *Cxcl5* in Dynabeads-isolated neutrophils (*Figure 3B*). Together, the analyses confirm that the up-regulation of inflammatory genes by E2B occurs mostly in neutrophils.

Next, we investigated the mechanism of estrogen-induced neutrophils infiltration. S100A8, S100A9 are known neutrophil chemoattractants that promote neutrophil migration during inflammation (*Wang et al., 2018*). S100A8 and S100A9 are small calcium-binding proteins that activate calcium-dependent signalling through receptor for advanced glycation end products (RAGE) or toll-like

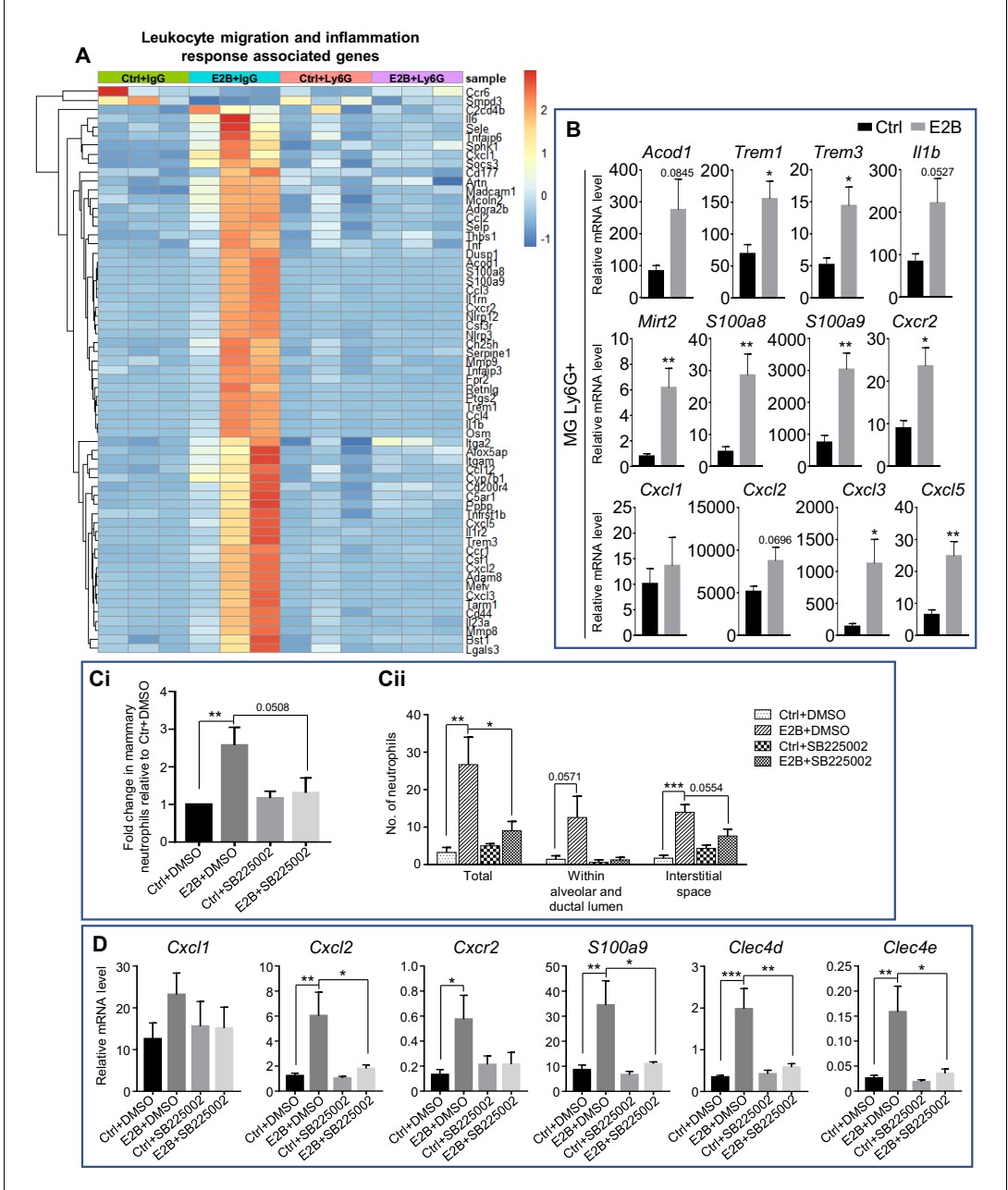

**Figure 3.** Estrogen-induced *Cxcr2* signalling in neutrophils plays a key role in neutrophil infiltration into the involuting mammary gland. (A) Heatmap representation of estrogen-regulated genes associated to leukocyte migration and inflammation in neutrophils (≥3 and ≤ −3-fold); Experiment is conducted according to the description in *Figure 2*. (B) qPCR analysis of estrogen-regulated expression of *Acod1*, *Mirt2*, *Trem1*, *Trem3*, *S100a9*, *S100a8*, *Cxcl1*, *Cxcl2*, *Cxcl3*, *Cxcl5*, *Cxcr2*, and *Il1b* relative to *Gapdh* in isolated mammary neutrophils following treatment with or without E2B for 24 hr (Ctrl n = 5, E2B n = 5). (C-D) E2B-induced *Cxcr2* in neutrophils is critical for E2B-induced neutrophil infiltration. Mice at INV D1 were treated with Ctrl or E2B in the absence or presence of CXCR2 inhibitor SB225002 for 48 hr; (C) SB225002 reduces E2B-induced mammary neutrophil (CD45+ CD11b+ Ly6G +) by flow cytometry analysis (Ci) (Ctrl+DMSO n = 7, E2B+DMSO n = 7, Ctrl+SB225002 n = 7, E2B+SB225002 n = 6), and the number of infiltrated neutrophils in 20 mm² of mammary sections (Cii) (Ctrl+DMSO n = 4, E2B+DMSO n = 3, Ctrl+SB225002 n = 3, E2B+SB225002 n = 3); (D) SB225002 reduces E2B-induced *Cxcl2*, *Cxcr2*, *S100a9*, *Clec4d*, and *Clec4e* expression in the involuting gland (Ctrl+DMSO n = 7, E2B+DMSO n = 6, Ctrl +SB225002 n = 7, E2B+SB225002 n = 5). Data represented as mean ± SEM.

The online version of this article includes the following figure supplement(s) for figure 3:

**Figure supplement 1.** Putative S100A9 inhibitor Paquinimod promotes neutrophil infiltration.

**Figure supplement 2.** 1H NMR Spectrum of the synthesized Paquinimod (PAQ): 1H NMR (CDCl3, 400 MHz) δ 7.43 (1H, dd, J = 8.6, 7.4 Hz), 7.29 – 7.10 (5H, m), 7.03 (2H, dd, J = 8.1, 5.5 Hz), 4.00 (2H, q, J = 7.1 Hz), 3.28 – 3.23 (3H+2H m), 1.30 (3H, t, J = 7.4 Hz), 1.23 (3H, t, J = 7.1 Hz).

receptor 4 (TLR4). Paquinimod (PAQ) is a derivative of quinoline-3-carboxamide that has been shown to inhibit S100A9 activity through binding with S100A9 (*Bengtsson et al., 2012*; *Björk et al., 2009*). The binding inhibits S100A9 dimerization or the formation of heterodimer with S100A8, thereby preventing the activation of RAGE or TLR4. To evaluate the role of S100A9 in estrogen-induced neutrophil infiltration, PAQ was synthesized following the reported method (*Jönsson et al., 2004*) and tested whether it inhibits E2B-induced neutrophil infiltration by treating OVX mice with E2B or E2B +PAQ at INV D1 for 48 hr in a pilot experiment. Flow cytometry analysis (*Figure 3—figure supplement 1A*) showed that E2B+PAQ treatment resulted in an increase in mammary neutrophils (CD45+ CD11b+ Ly6G+, p=0.0238) and monocytes (CD45+ CD11b+ Ly6C$^{hi}$, p=0.0242) when compared to E2B treated samples. Consistently, qPCR analysis also revealed a significant increase in pro-inflammatory markers such as *Cxcl2* (p=0.0124), *S100a8* (p=0.0233), *S100a9* (p=0.0161), and C-type lectin domain family 4, member e (*Clec4e*; p=0.0186) (*Figure 3—figure supplement 1C*). As it is not clear if PAQ has a yet to be characterized functional property that induces neutrophil infiltration, we cannot conclude from this experiment whether S100A8/A9 were significantly involved in the E2B-induced neutrophil infiltration.

CXCR2 is known to be important for neutrophil infiltration in a number of experimental settings (*Aziz et al., 2012*; *Alves-Filho et al., 2010*; *Cacalano et al., 1994*). Since *Cxcr2* and all its ligands were significantly up-regulated by E2B, we explored the involvement of CXCR2 signalling in E2B-induced neutrophil recruitment. OVX mice were treated with Ctrl or E2B at INV D1 for 48 hr in the presence of vehicle control DMSO, or CXCR2 antagonist SB225002 (*White et al., 1998*). To take into consideration of the large variations between experiments, the percentages of infiltrated mammary neutrophils (CD45+ CD11b+ Ly6G+) were presented as fold change over the Ctrl+DMSO group. As shown in *Figure 3Ci*, E2B treatment alone without the antagonist (E2B+DMSO) led to 2.57-fold increase (p=0.0082) in mammary neutrophils as compared to the Ctrl+DMSO. E2B +SB225002 treatment caused a 2.26-fold reduction in mammary neutrophils when compared to the E2B+DMSO group (p=0.0508). Quantification of the number of infiltrated mammary neutrophils within a 20 mm$^2$ area of stained mammary tissue also showed a significant reduction of neutrophil infiltration within mammary tissue (*Figure 3Cii*, p=0.0369). qPCR analysis of the mammary tissue samples showed that the E2B-induced pro-inflammatory markers such as *Cxcl2*, *S100a9*, *Clec4d*, and *Clec4e* were all significantly reduced with E2B+SB225002 treatment compared with E2B+DMSO treatment (*Figure 3D*, *Cxcl2*, p=0.0279; *S100a9*, p=0.0199; *Clec4d*, p=0.0051; *Clec4e*, p=0.0183). This is consistent with SB225002-induced inhibition of neutrophil infiltration because these genes are estrogen-induced in neutrophils. Collectively, the data showed that the E2B-induced activation of CXCR2 signalling and the up-regulation of CXCR2 ligands *Cxcl2*, *Cxcl3*, *Cxcl5* in neutrophils plays a pivotal role in estrogen-induced neutrophil recruitment.

## Estrogen-induced adipocyte repopulation is associated with the induction of adipogenic and tissue remodelling genes through neutrophils

Mammary adipocytes regress during pregnancy and lactation as the mammary epithelial proliferate and differentiate to fill up the mammary fat pad. Adipocytes expansion or repopulation is a hallmark of post-lactational mammary involution. *Figure 1* shows that estrogen significantly induces adipocyte repopulation during mammary involution, and neutrophil depletion attenuates it. We also determined the effect of neutrophil depletion by the Ly6G antibody on adipocytes repopulation in the intact (non-OVX) mice during mammary involution. Mice were given daily injection of Ly6G antibody from INV D1 for 2 or 3 days. Neutrophil depletion efficiency at INV D3 and D4 were ~ 95%. It was also shown that the infiltrated neutrophils at INV D4 is ~ 8 times of that at D3 (*Figure 4A*), indicating an increase of infiltrated neutrophils as involution proceeded. During sterile inflammation, infiltrated neutrophils was known to recruit monocytes from the circulation (*Shen et al., 2013*). In our study, neutrophils depletion also reduced mammary monocytes at both D3 and D4, although it is statistically not significant (*Figure 4B*). At INV D3, neutrophil depletion showed no effect on adipocytes repopulation (*Figure 4C*, p=0.3737 and *Figure 4—figure supplement 1*). However, there was a significant reduction in adipocytes repopulation with neutrophil depletion at INV D4 based on perilipin immunostaining (*Figure 4C*, p=0.0077, and *Figure 4—figure supplement 1*). The data suggest that neutrophils involvement in adipocytes repopulation during mammary involution is limited to a specific time window when the levels of neutrophil infiltration is high. GO over-representation analysis

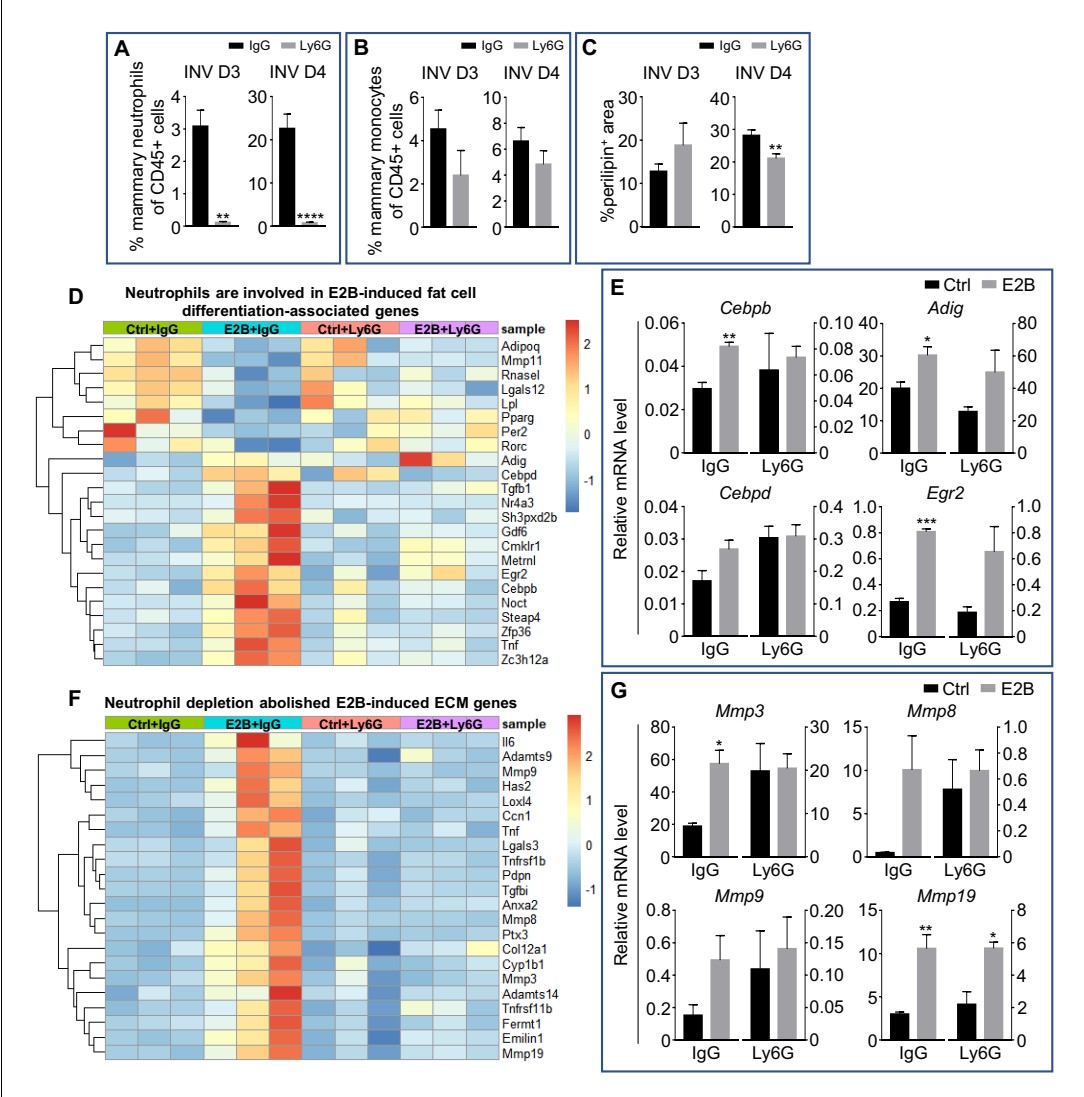

**Figure 4.** Estrogen-induced adipocyte repopulation is associated with induction of adipogenic and tissue remodelling genes in neutrophils. (A-C) Non-OVX mice were treated daily with either anti-Ly6G antibody (Ly6G) or isotype control (IgG) from 24h post-weaning (INV D1); (A) Flow cytometry analysis of mammary neutrophils (CD45+ CD11b+ Gr1$^{hi}$) from mice treated with IgG or Ly6G; (B) Flow cytometry analysis of mammary monocytes (CD45+ CD11b+ Ly6C$^{hi}$) from mice treated with IgG or Ly6G; (C) Quantification of percentage of perilipin stained area at INV D3 (IgG n=3, Ly6G n=4) and INV D4 (IgG n=10, Ly6G n=9). (D) Heatmap representation of genes associated to fat cell differentiation identified from the GO over-representation analysis ($\geq$ 1.5 and $\leq$ -1.5-fold); Experiment is conducted according to the description in *Figure 2*; (E) Gene expression of adipogenesis genes *Adig, Egr2, Cebpb,* and *Cebpd* relative to *Rplp0* by qPCR analysis. (F) Heatmap representation of genes associated to extracellular matrix organization identified from the GO over-representation analysis ($\geq$ 2 and $\leq$ -2-fold); Section of heatmap replotted from *Figure 4—figure supplement 2*; Experiment is conducted according to the description in *Figure 2*; (G) Gene expression of tissue remodelling genes *Mmp3, Mmp8, Mmp9,* and *Mmp19* relative to *Rplp0* by qPCR analysis (Ctrl+IgG n=3, Ctrl+Ly6G n=3, E2B+IgG n=3, E2B+Ly6G n=3). Data represented as mean ± SEM.

The online version of this article includes the following figure supplement(s) for figure 4:

**Figure supplement 1.** Neutrophil depletion transiently reduces E2B-induced adipocyte repopulation during mammary involution.

**Figure supplement 2.** 50% of estrogen-regulated ECM genes are abolished by neutrophil depletion.

**Figure supplement 3.** Estrogen regulation of lipid metabolism during post-weaning mammary involution were attenuated with neutrophil depletion.

of RNA-Seq data indicates that E2B regulated genes that are associated with fat cell differentiation. Neutrophils depletion attenuated most of E2B-induced genes in fat cell differentiation category (*Figure 4D*). Intriguingly, some of these E2B-regulated changes indicate anti-adipogenic effect, whereas others appear pro-adipogenic. For example, E2B down-regulated the expression of *Adipoq, Pparg,* and *Lpl* independent of neutrophils, all of which are positive regulators of adipogenesis.

On the other hand, E2B also up-regulated some of the upstream regulators of adipogenesis, such as early growth response 2 (*Egr2*), adipogenin (*Adig*), CCAAT/enhancer binding protein (C/EBP), beta (*Cebpb*), and CCAAT/enhancer binding protein (C/EBP), delta (*Cebpd*), all of which were validated by qPCR analysis (*Figure 4E*). Neutrophil depletion abolished the up-regulation of *Egr2*, *Adig* and *Cebp/b*. Interestingly, although *Cebpb* and *Cebpd* are no longer upregulated by E2B in neutrophils depleted samples, the relative expressions of *Cebpb* and *Cebpd* were increased upon neutrophil depletion (*Figure 4E*). This seems to suggest that neutrophils depletion exerted positive influence on the expression of *Cebpb* and *Cebpd* in mammary tissue and this would limit a further increase in response to E2B.

ECM remodelling also plays a key role in adipocyte repopulation during mammary involution (*Alexander et al., 2001*). GO over-representation analysis showed 43 ECM organization associated genes with fold regulation ≥ 2 fold (*Figure 4—figure supplement 2*). 22 of the 43 ECM-related genes were induced by E2B, and their inductions were abolished with neutrophil depletion (*Figure 4F*). Since matrix metallopeptidases (MMPs) are strongly induced in obese adipose tissue and modulate during adipocytes differentiation (*Chavey et al., 2003*), the expression of *Mmp3*, *8*, *9*, and *19* were validated by qPCR. All four Mmps were upregulated by E2B in the IgG group (*Figure 4G*), although the induction of *Mmp8* and *Mmp9* was not statistically significant due to one outlier in the samples. This was likely due to variations in neutrophil numbers in these samples as the expression of MMP8 was reported to occur mainly in neutrophils (*Owen et al., 2004*). The E2B-induced up-regulation of *Mmp3*, *Mmp8* and *Mmp9* was abolished or attenuated significantly with neutrophil depletion (*Figure 4G*), suggesting that E2B upregulated these *Mmp*s in neutrophils or in other cell types due to the activity of neutrophils.

E2B also regulated genes related to lipid metabolic process (*Figure 4—figure supplement 3*). Consistently, the regulations of most of these genes were also attenuated by neutrophil depletion. The top five up-regulated genes based on fold induction are *Il1b*, *Ptgs2* (coding for COX-2), *Nr4a3*, *Fpr2* and *Hcar2*. *Il1b* has been reported to inhibit lipolysis (*Bing, 2015*), whereas *Ptgs2*, *Fpr2*, and *Hcar2* are known to promote lipolysis and adipogenesis (*Ghoshal et al., 2011*; *Chen et al., 2019*; *Ren et al., 2009*).

Taken together, the study suggests that E2B regulated both pro and anti-adipogenic genes. Most of these effects are attenuated by neutrophil depletion. Thus, neutrophils play a role in E2B stimulation of adipocytes repopulation. However, other cell types may also be involved. For example, E2B also induced infiltration of monocytes which share some common properties with macrophages, and macrophages have been shown to be critical for mammary adipocytes repopulation (*O'Brien et al., 2012*).

## Estrogen accelerates lysosomal-mediated programmed cell death during involution

Mammary epithelial cells undergo LM-PCD during the early stage of involution as lysosomes in the mammary cells undergo membrane permeabilization, releasing lysosomal proteases into the cytosol, triggering apoptosis independent of executioner caspases such as caspase 3, 6, and 7 (*Sargeant et al., 2014*; *Kreuzaler et al., 2011*; *Watson and Kreuzaler, 2011*). Signal transducer and activator of transcription 3 (STAT3) has been shown to be a regulator for LM-PCD during involution by inducing the expression of lysosomal proteases cathepsin B (CTSB) and L (CTSL), while down-regulating their endogenous inhibitor Spi2A (*Kreuzaler et al., 2011*).

Despite its well-known function in stimulating mammary growth, estrogen significantly enhanced cell death during the acute phase of mammary involution (*Figure 1A and B*, p<0.05). Since both RNA-Seq analysis and qPCR demonstrated E2B up-regulation of *Ctsb* expression (*Figure 5A*, IgG, p=0.0163; Ly6G, p=0.0182, and *Figure 5—figure supplement 1*), we evaluated the protein levels of cathepsins. Indeed, the pro and active form (sc, single-chain) of CTSB were significantly up-regulated (pro-CTSB, p=0.0358; sc-CTSB, p=0.0101) in the E2B-treated mammary gland. Furthermore, the sc-form of cathepsin D (CTSD) and CTSL were also significantly up-regulated by E2B (*Figure 5B*, sc-CTSD, p=0.0222; sc-CTSL, p=0.0385), although the levels of pro-CTSD and CTSL were unchanged. Since E2B had no effect on the expression of *Ctsd* and *Ctsl* (data not shown), the increase of the sc-form of CTSD and CTSL can be explained by the up-regulation of CTSB which catalyses the removal of the N-terminal propeptide from itself and from CTSD and CTSL, leading to an increase in active forms of cathepsins B, D, and L (*Mach et al., 1994*; *Ménard et al., 1998*). Consistent with the lack

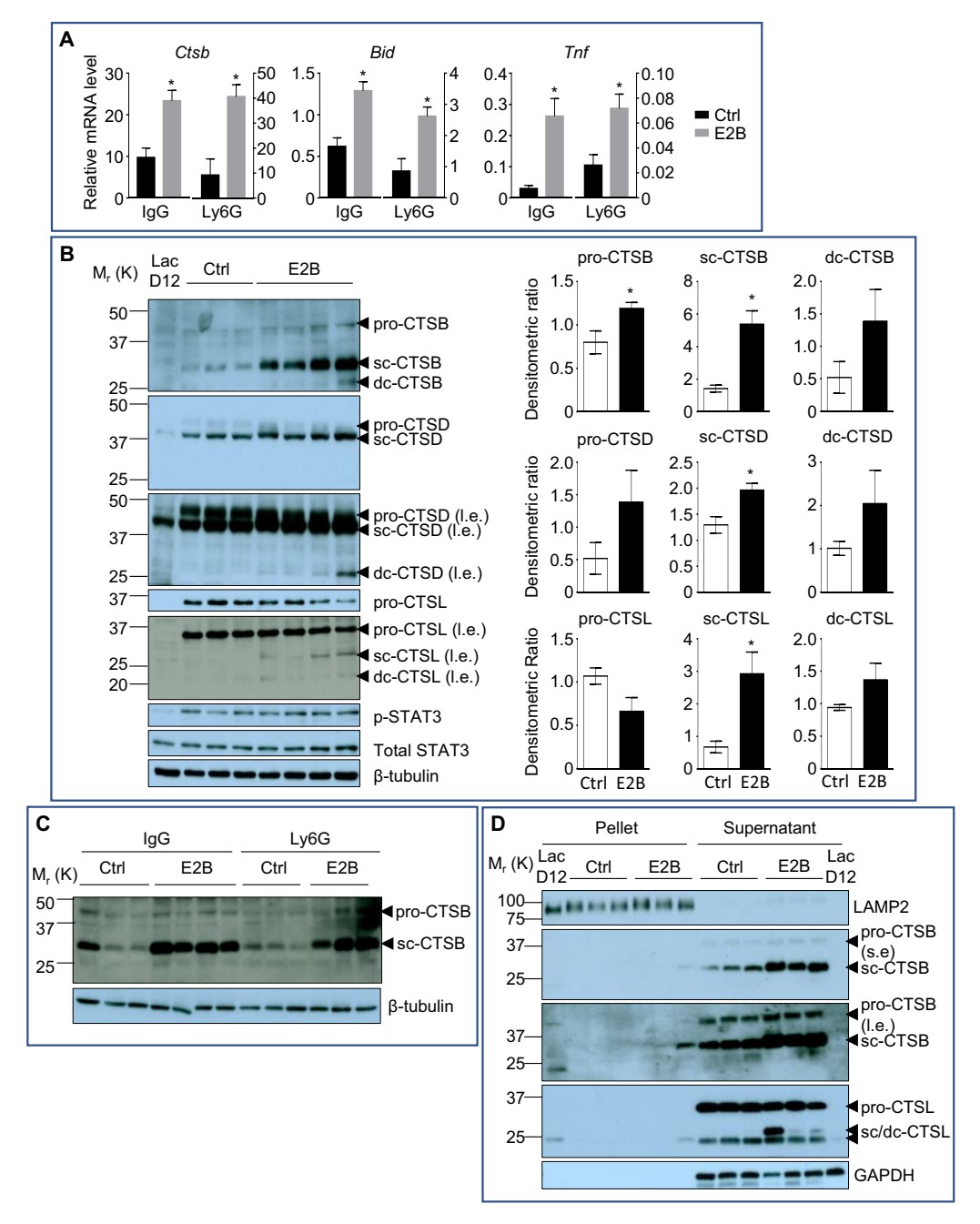

**Figure 5.** Estrogen stimulates the activity of lysosomal proteases that are critical for LM-PCD. (**A**) qPCR validation of E2B-induced expression of *Bid*, *Ctsb*, and *Tnf* relative to *Rplp0* identified from DESeq2 analysis (*Figure 5—figure supplement 1*) (Ctrl+IgG n = 3, Ctrl+Ly6G n = 3, E2B+IgG n = 3, E2B +Ly6G n = 3). Mice on INV D1 were treated with Ctrl or E2B for 48 hr before MG were collected for analysis; (**B**) Western blots of cathepsin B, D and L proteins (sc, single-chain; dc, heavy chain of the double-chain form) in mammary tissue of 48 hr treatment (Ctrl n = 3, E2B n = 4). (**C**) Western blotting analysis shows that depletion of neutrophils did not affect estrogen-induced increase of single-chain (sc) and double-chain (dc) forms of CTSB (Ctrl+IgG n = 3, E2B+IgG n = 4, Ctrl+Ly6G n = 3, E2B+Ly6G n = 3). (**D**) Effect of E2B on protein levels of lysosomal and cytosolic CTSB and CTSL proteins after subcellular fractionation. LAMP2 is used as a lysosomal marker (s.e, short exposure; l.e., long exposure) (Ctrl n = 3, E2B n = 3). Data are presented as Mean ± SEM.

The online version of this article includes the following figure supplement(s) for figure 5:

**Figure supplement 1.** Heatmap representation of estrogen-regulated genes associated with cell death from the GO over-representation analysis.

**Figure supplement 2.** Estrogen-induced genes associated with cell proliferation are regulated in both mammary gland and neutrophil population.

of neutrophil influence on E2B-induced cell death, neutrophil depletion had no effect on E2B-induced increase of active form of CTSB (*Figure 5C*).

E2B also significantly induced the expression of tumor necrosis factor (*Tnf*) (*Figure 5A*, IgG, p=0.0161; Ly6G, p=0.0343, and *Figure 5—figure supplement 1*), which is a known upstream activator of STAT3. However, E2B did not affect the levels of phosphorylated and total STAT3 (*Figure 5B*). This suggests that the up-regulation of *Ctsb* expression by E2B is a direct event independent of STAT3 activation. Furthermore, E2B also up-regulated the expression of BH3 interacting domain death agonist (*Bid*), independent of neutrophils (*Figure 5A*, IgG, p=0.0119; Ly6G, p=0.0248, and *Figure 5—figure supplement 1*). *Bid* is a well-established pro-apoptotic marker, and its overexpression has been reported to promote cell death (*Fukazawa et al., 2003*; *Li et al., 2012*).

Physiological LM-PCD during mammary involution is associated with the leakage of lysosomal cathepsins into the cytosol. We further tested if estrogen-induced increase of active CTSB and CTSL protein is associated with an increase of cytosolic cathepsins by lysosome and cytosol fractionation. Lysosome-associated membrane protein 2 (LAMP2) was used as a lysosomal marker (*Figure 5D*). Consistently, E2B-induced increases of the sc-form of CTSB and CTSL occurred mostly in the cytosolic fraction (*Figure 5D*). These findings further support the notion that E2B promotes mammary cell death by increasing the activity of lysosomal proteases CTSB, CTSD, and CTSL under the cellular condition of mammary involution.

Estrogen-induced cell death under physiological condition is hitherto unreported. We hypothesized that pro-inflammatory condition may prime estrogen-induced cell death during the acute phase of mammary involution. As wild-type MCF7 does not express caspase-3 (*Jänicke, 2009*), the hypothesis was tested in an in vitro model using MCF7-caspase3(+) breast cancer cell line (*Jänicke et al., 1998*). The idea was to recapitulate estrogen-induced cell death during mammary involution using the pro-inflammatory cytokine TNFα. MCF7-caspase3(+) cells were first treated with TNFα or vehicle control for 1 hr. This was followed by treatment with 17β-estradiol (E2) or vehicle control for 24 hr before the cells were stained with propidium iodide (PI) for analysis with flow cytometer. Cells undergoing programmed cell death exhibit increased membrane permeability and hence will stain positive for PI. As shown in *Figure 6A*, increasing concentration of TNFα from 2 ng/ml to 10 ng/ml led to a dose-dependent increase in the percentage of dead cells (p<0.01). As expected, E2B treatment alone did not cause cell death. However, E2B+TNFα treatment significantly enhanced the percentage of dead cells compared to TNFα treatment alone at doses of 5 ng/ml (p=0.0104) and 10 ng/ml (p=0.0007). Expectedly, TNFα induced increases in p-STAT3 and cleaved poly-ADP ribose polymerase (PARP) protein as compared to the vehicle-treated controls (*Figure 6B*). E2B+TNFα increased the levels of p-STAT3 and cleaved PARP significantly compared to TNFα treatment alone (*Figure 6B* and C, p-STAT3, p=0.0404; cleaved PARP, p<0.0001). These molecular changes are consistent with a greater level of cell death induced by the combined treatment of E2B and TNFα. Together, the in vitro data support the notion that pro-inflammatory cytokines such as TNFα primes the death-inducing effect of estrogen. However, the increased p-STAT3 is not associated with increases of active CTSD and CTSB (data not shown), suggesting that estrogen-induced cell death in the presence of TNFα may not be via LM-PCD in the MCF7-caspase3(+) cells.

## Estrogen remains a mitogenic hormone during mammary involution

Despite its effect on mammary cell death, estrogen retains its function as a mitogen as indicated by E2B-induced expression of pro-growth genes in the RNA-Seq analysis. The up-regulation of amphiregulin (*Areg*), epiregulin (*Ereg*), and myelocytomatosis oncogene (*c-Myc*) were validated by qPCR in the IgG group (*Figure 5—figure supplement 2*, *Areg*, p=0.0448; *Ereg*, p=0.0435; *c-Myc*, p=0.0063) and the E2B's effects were largely unaffected by neutrophil depletion. These data provide evidence that during post-weaning mammary involution, following the massive cell death events, estrogen promotes the regeneration of the mammary gland.

## The effect of estrogen in age-matched nulliparous mammary tissue is distinct from that undergoing mammary involution

To confirm that the effect of estrogen on neutrophils and on gene regulation during mammary involution does not occur similarly in the post-pubertal mammary gland, OVX nulliparous mice were

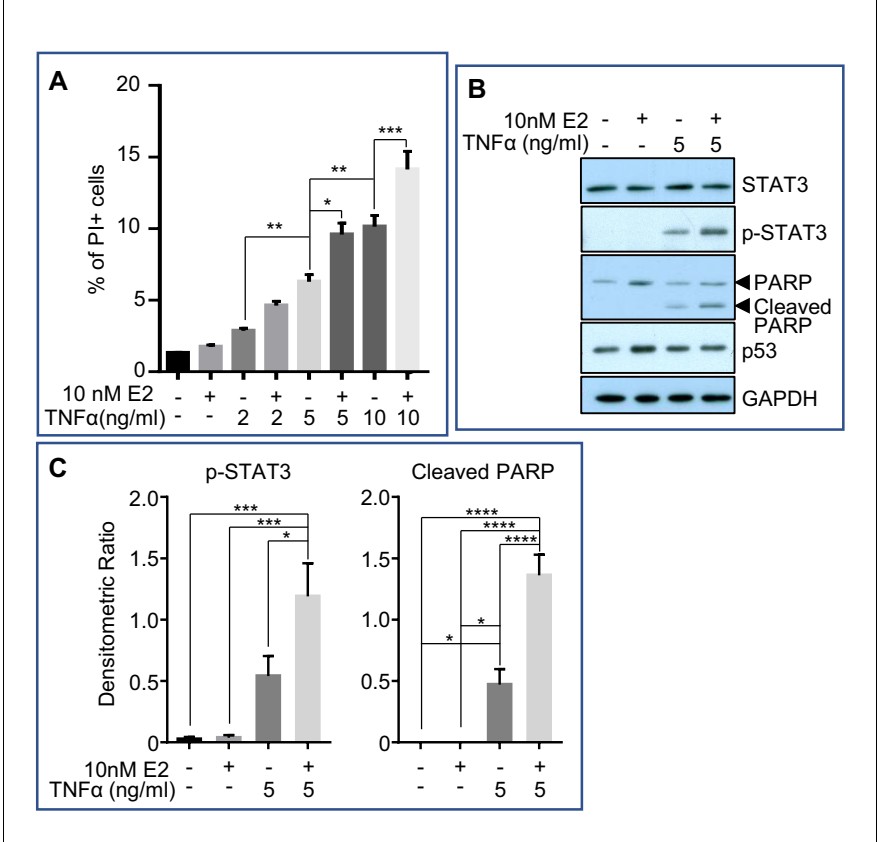

**Figure 6.** Estrogen accelerates TNFα-induced cell death in vitro. MCF7-caspase3(+) cells were treated with either vehicle control (1xPBS) or TNFα of varying concentrations. An hour later, cells were treated with either vehicle control (0.01% ethanol) or 10nM 17β-estradiol (E2) for 24h, after which they were collected for analysis. (**A**) Flow cytometry analysis for the percentage of propodium iodide (PI)-positive cells (dead cells) after treatment (4 independent experiments with triplicates for each group). (**B**) Representative western blotting analysis of various proteins from the treated cells. (**C**) Densitometric analysis of protein expressions normalized against GAPDH (3 independent experiments with duplicates for each group). Data represented as mean ± SEM.

given isotype IgG the day prior to E2B treatment in order to match the IgG treatment in the involuting mice. 24 hr later, these animals were given either vehicle control or E2B for 24 hr. Flow cytometry analysis of the mammary gland showed that E2B treatment resulted in a 56.33% decrease in mammary neutrophils (CD45+ CD11b+ Gr1[hi]) (*Figure 7A*, p=0.0349). E2B had no significant effect on the percentage of mammary monocytes (CD45+ CD11b+ Ly6C[hi], p=0.2965). qPCR analysis was also performed to investigate the effect of E2B on the expression of pro-inflammatory markers and cell death-related genes. E2B significantly down-regulated *S100a9* (p=0.0095), *Cxcl2* (p=0.0382), and *Cxcr2* (p=0.0236) while had no effect on the expression of *S100a8*, *Clec4d*, *Il1b*, and *Trem3* (*Figure 7C*). The expression of *Clec4e*, *Mirt2*, and *Trem1* were also analyzed but had no amplification in the qPCR reactions. These observations contrast with that in the involuting mammary gland where the expression of all these pro-inflammatory genes were up-regulated with E2B treatment. As for the cell death-related genes, E2B up-regulated *Ctsb* (p=0.0166), consistent with the understanding that *Ctsb* is an ER target gene (*Lee and Choi, 2013*). However, E2B had no effect on the expression of *Bid* while *Tnf* displays no amplification in the qPCR reaction due to the low level of expression (*Figure 7D*). Hence, unlike its effect during mammary involution, E2B exerts an anti-inflammatory effect on the mammary gland of nulliparous mice.

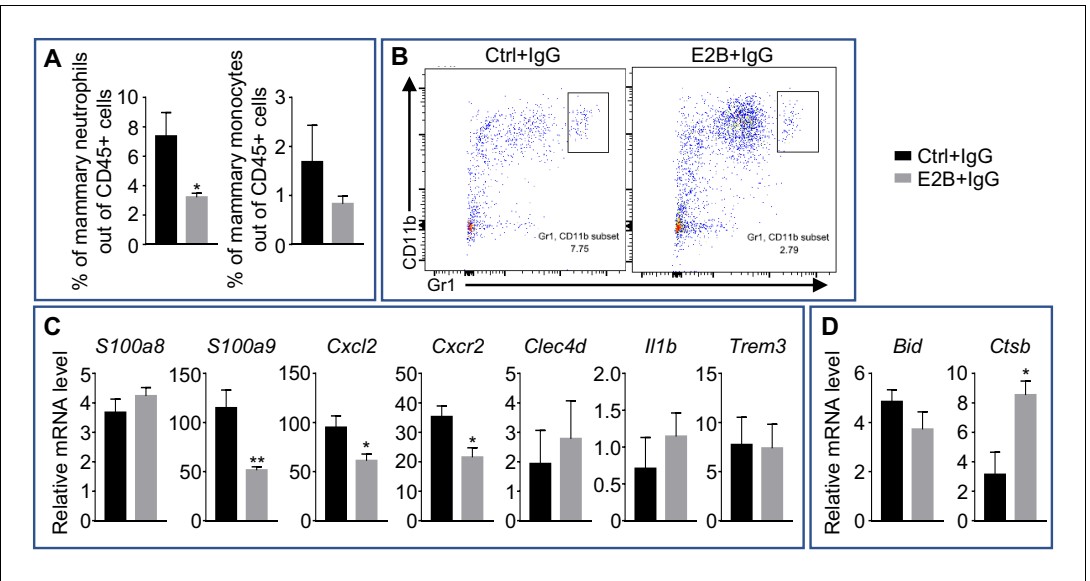

**Figure 7.** Estrogen regulation of inflammatory and apoptotic genes in nulliparous mammary tissue. OVX nulliparous mice were treated with isotype control (IgG). 24 hr later, they were treated with either Ctrl or E2B for 24 hr. (**A**) Flow cytometry analysis of mammary neutrophils (CD45+ CD11b+ Gr1^hi) and monocytes (CD45+ CD11b + Ly6C^hi). (**B**) Representative flow cytometry dot plot for the percentage of neutrophils in the MG. (**C**) Gene expression of pro-inflammatory genes *S100a8*, *S100a9*, *Cxcl2*, *Cxcr2*, *Clec4d*, *Il1b*, and *Trem3* relative to *Rplp0* by qPCR. (**D**) Gene expression of cell death associated genes *Bid* and *Ctsb* relative to *Rplp0* by qPCR. Ctrl+IgG n = 5, E2B+IgG n = 5. All data are presented as mean ± SEM.

## Estrogenic stimulation of mammary inflammation and mammary cell death is mediated by ERα

Mammary gland and neutrophils express both ERα and ERβ. To elucidate which of these ER subtypes mediates the effect of E2B on the various process of mammary involution, we evaluated the effect of ERα-specific agonist, (4,4′,4′′-(4-Propyl-[1H]-pyrazole-1,3,5-triyl) trisphenol (PPT) (*Stauffer et al., 2000*) and ERβ-specific agonist Diarylpropionitrile (DPN) (*Meyers et al., 2001*). Ovariectomized mice were treated with control vehicle, PPT, or DPN on INV D1. Mammary gland were collected for analysis after 48 hr's treatment. Similar to the effect of E2B, ERα agonist PPT evidently increased the expression of *Ctsb*, *Bid* and *Tnf* that are known to promote cell death (*Figure 8A*). PPT also consistently increased the levels of cleaved CTSB (sc-CTSB), cleaved CTSD (sc-CTSD), and cleaved CTSL (sc-CTSL and dc-CTSL) (*Figure 8B*), which are executioner proteases (*Foghsgaard et al., 2001*; *Luzio et al., 2007*). Unlike the effect of E2B which also increased pro-CTSB, PPT did not seem to increase pro-CTSB. We suspect that the PPT-induced increase of pro-CTSB was already converted to the cleaved form as the pharmacodynamics of E2B and PPT are likely different. In contrast, ERβ-specific agonist DPN did not have any effect on the gene expression of the death-inducing genes, or the activation of lysosomal proteases (*Figure 8A and B*). These observations indicate that ERα mediates E2B-induced LM-PCD during mammary involution.

We also analyzed a panel of genes known to be regulated by E2B through the activity of neutrophils. PPT, but not DPN treatment, significantly increased mammary expression of *Cxcl1*, *Cxcl2*, *Cxcl2*, and *Mirt2* (*Figure 8A*). This suggests that the regulatory effect of E2B on mammary inflammation and on neutrophil activity in particular is primarily mediated by ERα.

Similar to the effect of E2B, PPT, but not DPN increased the expression of adipogenesis-related genes, *Ptgs2* (*Cox-2*), *Adig*, and *Egr2* (*Figure 8A*). However, neither PPT nor DPN affected the expression of *Cebpb*. We also looked at their effects on *Mmps*. Both PPT and DPN did not affect the expression of *Mmp3* but increased the expression of *Mmp9*. On the other hand, PPT but not DPN enhanced the expression of *Mmp8*. We deduce from these observations that ERα is likely involved in E2B-induced adipocytes repopulation through the activity of neutrophils. It is possible

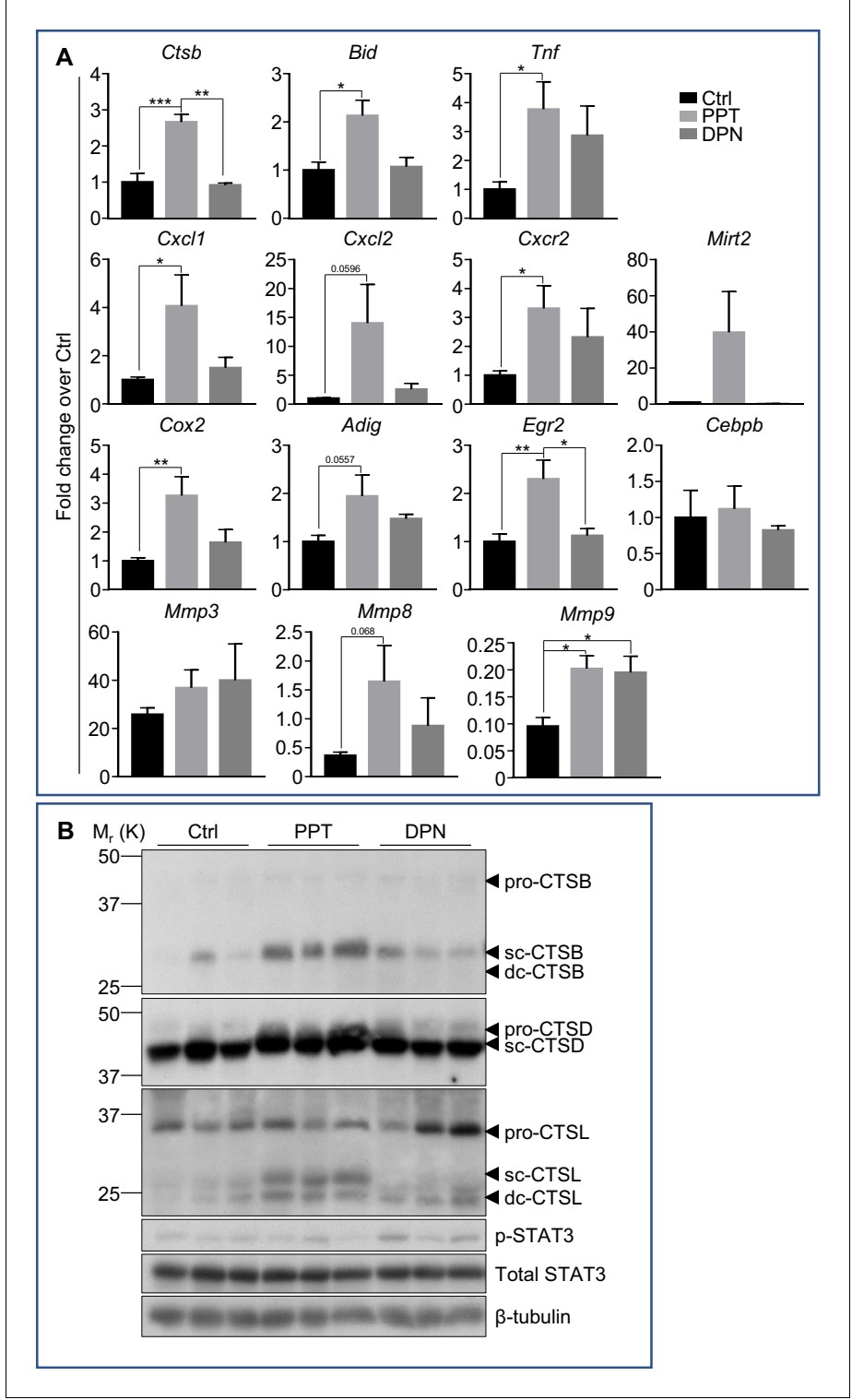

**Figure 8.** Estrogenic stimulations of neutrophil gene expression and mammary cell death are mediated by ERα. Mice at INV D1 was treated with vehicle control (Ctrl), ER-α agonist PPT, or ER-β agonist DPN for 48h before mammary gland were collected for analysis. (**A**) qPCR analysis of gene expression in mammary tissue. The results are expressed as fold change (Mean ± SEM) in response to PPT or DPN (Ctrl n=7, PPT n=5, DPN n=3). (**B**) Western blotting analysis of cathepsin B, D, L, and STAT3 proteins (sc, single-chain; dc, heavy chain of the double-chain form) in mammary tissue (Ctrl n=3, PPT n=3, DPN n=3).

The online version of this article includes the following figure supplement(s) for figure 8:

*Figure 8 continued on next page*

*Figure 8 continued*

**Figure supplement 1.** Expression of *Esr1* (ERα) is about 40 times higher than *Esr2* (ERβ) in mammary neutrophils during mammary involution.

that ERβ is negatively involved in the regulation of adipocytes repopulation because ERβ is known to repress the transcriptional activity of PPARγ and inhibit adipocyte differentiation (*Foryst-Ludwig et al., 2008*). But these effects may not involve neutrophils.

## Discussion

This is a comprehensive study of biological effects of estrogen on the acute phase of mammary involution. We report that estrogen accelerates mammary involution by exacerbating mammary inflammation, programmed mammary cell death, and adipocytes repopulation. These effects were found to be mediated through distinct mechanisms. First, the effect on mammary inflammation is primarily mediated through the activities of mammary neutrophils. RNA-Seq analysis revealed the remarkable extent of estrogen regulation of neutrophil activity. Neutrophil depletion eliminated 88% of estrogen-regulated genes in mammary tissue, even though neutrophils account for less than 5% and 1% of the total number of live cells in FACS analysis of estrogen-treated and control samples respectively. This 88% of the genes were regulated by estrogen either in neutrophils, or as a result of neutrophil activities on other cells. Functional analysis showed that neutrophils are the primary mediators of estrogen-induced inflammation and neutrophil infiltration. The remarkable impact of estrogen on neutrophil activities would conceivably have a significant influence on the mammary tissue microenvironment. This affirms the pivotal roles of neutrophils in mediating the pro-tumoral effect of estrogen on ER-negative tumor development in mammary tissue during involution (*Chung et al., 2017*). Second, estrogen promotes mammary LM-PCD independent of neutrophils by inducing the expression and activity of lysosomal cathepsins and other pro-apoptotic markers such as *Bid* and *Tnf*. It is well known that dying cells release signaling molecules to stimulate inflammation and recruitment of phagocytic cells for their clearance (*Rock and Kono, 2008*). Likewise, estrogen-induced cell death may aggravate inflammation, leading to a greater level of neutrophil recruitment. Third, neutrophils play an important role in estrogen-induced adipocytes repopulation as neutrophils depletion attenuated adipocytes repopulation. Notably, all these effects of estrogen are unique to the mammary tissue during post-weaning involution, signifying the plasticity of estrogen action depending on the tissue microenvironment.

### Plasticity of estrogen action on neutrophils

In sharp contrast to the effect on neutrophils during mammary involution, estrogen reduced mammary neutrophil infiltration in age-matched nulliparous mice. Estrogen-regulated expression of cytokines such as *S100a9*, *Cxcl2*, and *Cxcr2* in nulliparous mice also occurs in the opposite direction as that in mice undergoing mammary involution. The observation in nulliparous mice is consistent with reports that estrogen inhibits inflammation in obesity-induced mammary inflammation (*Bhardwaj et al., 2015*), and in *Staphylococcus aureus* infected bovine mammary epithelial cells (*Medina-Estrada et al., 2016*). To our knowledge, this is the first evidence of the plasticity of estrogen action on neutrophils that is shaped by the tissue microenvironment in an in vivo model. This suggests that there are fundamental differences in the cistrome of mammary neutrophils between these two states.

   Neutrophils are known to express ERα, ERβ, and GPER30 (*Molero et al., 2002*). ERα and ERβ are members of the nuclear receptor superfamily of transcription factors, whereas GPER30 is a G protein-coupled membrane receptor. Several lines of evidence suggest that the effect of estrogen on mammary neutrophils during mammary involution is mediated by ERα. Firstly, the relative expression of *Esr1* (ERα) in mammary neutrophils during involution is approximately 40 times that of *Esr2* (ERβ) (*Figure 8—figure supplement 1*). Secondly, ERα-specific agonist PPT elicited similar changes as E2B on the expression of a panel of inflammatory and adipogenesis-related genes in neutrophil (*Figure 8*). In contrast, ERβ-specific agonist DPN, had no effect. ERα has also been reported to mediate estrogen-induced neutrophil migration in the uterus through ERα phosphorylation at serine 216

(*Shindo et al., 2013*), and ERα mediated the effect of estrogen on myeloid-derived suppressor cells, which are mostly granulocytic cells, in stimulating tumor development in mice model (*Svoronos et al., 2017*). Thus, ERα plays a major part in mediating estrogen regulation of neutrophils activity during mammary involution.

Immune cells are notably plastic in their ability to adapt to changes in their environment. There has been growing evidence of phenotypic heterogeneity and functional plasticity in neutrophils, which has been the subject of comprehensive reviews (*Ng et al., 2019*; *Silvestre-Roig et al., 2019*). How neutrophils evolve into different phenotype with distinct function is still unclear. Ng et al suggested two non-mutually exclusive mechanisms: intrinsic heterogeneity in the bone marrow and blood, and extrinsic heterogeneity inducible by local or systemic factors (*Ng et al., 2019*). The latter would conceivably apply to neutrophils in tissues, resident or conditionally infiltrated, that evolve into specialized phenotypes in order to meet the developmental demand. The distinct transcriptional responses to estrogen in infiltrated mammary neutrophils between post-weaning mice and nulliparous mice reflects this functional plasticity during development. Given that estrogen regulation of gene transcription is mediated by ERα, we postulate that signaling molecules in inflammatory mammary tissue remodel the epigenetic landscape of neutrophils so as to modify ERα cistrome. It has been reported that neutrophils undergo dynamic changes in their epigenome during development (*Grassi et al., 2018*; *Rönnerblad et al., 2014*). Under mature state, neutrophils respond to environmental cues with epigenetic modification and transcription changes. For example, the inactive IL-6 genomic locus in human neutrophils undergoes chromatin remodeling in response to ligands for TLR8, leading to IL-6 secretion (*Zimmermann et al., 2015*). It is known that ERα cistrome is reprogrammable and TNFα has been shown to reshape the genomic action of estrogen in breast cancer cells through redistribution of NF-kB and pioneer factor FOXA1 binding across the genome (*Frasor et al., 2009*; *Franco et al., 2015*). Thus, the inflammatory microenvironment in mammary tissue during involution may trigger a global shift of ER-cistrome in mammary neutrophils so as to modify estrogen response that is distinct from that in nulliparous mice. The mammary involution model is thus useful for the elucidation of mechanisms that drive epigenetic and phenotypic changes in neutrophils.

It should be recognized that ERβ has been reported to play a role in neutrophil regulation of tumor biology. Selective ERβ agonist LY500307 was found to reduce lung metastasis of triple-negative breast cancer cells (*Zhao et al., 2018*). This was associated with significant increase of infiltration of neutrophils in the lung. However, this was mediated by the up-regulation of IL-1β in cancer cells which promotes neutrophil infiltration to the metastatic niche. Whether the ERβ agonist regulated the phenotypes of neutrophils directly is not clear.

## Estrogen promotes CXCR2 signalling to elicit neutrophil infiltration

The involvement of CXCR2 signalling in neutrophil infiltration has been widely reported. In an acute lung injury model, *Cxcr2* gene deletion abolished hyperoxia-induced neutrophil accumulation (*Sue et al., 2004*). In studies of reperfusion injury, inhibition of CXCR2 with repertaxin or anti-CXCR2 antibodies led to the reduction of neutrophils accumulation (*Cugini et al., 2005*; *Belperio et al., 2005*; *Bertini et al., 2004*). The CXCR2 antagonist SB225002 has also been reported to reduce neutrophil recruitment and pro-inflammatory factor expression in LPS-induced acute lung injury (*Cao et al., 2018*). Using CXCR2 antagonist SB225002, this study identified CXCR2 signalling as a major pathway for estrogen to induce neutrophil infiltration in mammary tissue during mammary involution (*Figure 3C and D*). This is likely mediated by the significant upregulation of *Cxcr2* and its ligands *Cxcl2*, *Cxcl3*, and *Cxcl5* in neutrophils (*Figure 3B*). It is conceivable that the recruitment of neutrophils involves an autocrine loop, in which estrogen induces neutrophil release of CXCR2 ligands which activate CXCR2 signalling in neutrophils.

Additionally, a number of the estrogen-regulated genes through neutrophils are also linked to chemotaxis, leukocytes adhesion and migration. CXCR2 signalling likely cooperates with other estrogen-induced chemotactic factors. *Trem1* and *Trem3* are among the top estrogen-regulated genes in neutrophils (*Figure 3A*). TREM1 was first identified to be selectively expressed on neutrophils and monocytes (*Bouchon et al., 2000*). TREM3 is highly homologous to TREM1 and is believed to have overlapping function. TREM1/3-deficient mice displayed impaired neutrophil trans-epithelial infiltration into the lung when challenged with *P. aeruginosa* (*Klesney-Tait et al., 2013*). TREM1 was also known to amplify inflammation, as TREM1 overactivation with activating antibodies following LPS

treatment led to the up-regulation of cytokines such as TNFα, MCP-1, and IL8 (*Bouchon et al., 2000*). It is plausible that *Trem1/3* up-regulation also plays a role in estrogen-induced neutrophil infiltration.

## Estrogen-stimulated neutrophils play a part in adipocytes repopulation

Estrogen is traditionally known to promote metabolism and inhibit adipogenesis (*Chen and Brown, 1793*). It has been reported recently that ERα signalling is required for the adipose progenitor identity and the commitment of white fat cell lineage (*Lapid et al., 2014*). The present study provides the first evidence that estrogen stimulates adipocytes repopulation during mammary involution. This effect likely involved neutrophils because neutrophils depletion attenuated estrogen-induced adipocytes repopulation and the gene expression related to fat cell differentiation, lipid metabolism, and ECM remodelling (*Figure 1D*). It is to be noted that mammary macrophages and infiltrated monocytes likely play a role in estrogen-induced adipocytes repopulation. First, macrophages have been shown to be critical for adipocytes repopulation during mammary involution (*O'Brien et al., 2012*). Whilst we did not examine the effect of estrogen on mammary resident macrophages, estrogen has been reported to regulate gene expression of peritoneal macrophages involved in various processes including cell proliferation, immune response and wound healing (*Pepe et al., 2017*). Thus, estrogen may stimulate macrophage activity for adipocytes repopulation. Second, estrogen also induces infiltration of monocytes into mammary tissue (*Chung et al., 2017*), which share common properties with F4/80+ macrophages. Third, consistent with the understanding that infiltrated neutrophils recruit monocytes into the tissue during sterile inflammation (*Shen et al., 2013*), we observed that estrogen-induced neutrophil depletion reduced, to a small extent, infiltrating monocytes (*Figure 2— figure supplement 1B*). Taken together, estrogen-induced adipocytes repopulation is plausibly mediated by the activities of neutrophils, infiltrated monocytes as well as resident macrophages in the mammary gland.

## Estrogen promotes LM-PCD during mammary involution

Mammary involution in the first 48 hr is known as phase I (reversible phase) of mammary involution. In phase I, LM-PCD has been shown to be the major mechanism of mammary cell death, and it is independent of caspases 3, 6, and 7 (*Kreuzaler et al., 2011*). The present study provides the first evidence that estrogen treatment in the reversible phase of mammary involution accelerates cell death when there is ongoing LM-PCD. This is mediated by increased expression of *Ctsb*, and the cytosolic protein levels of active (cleaved) CTSB, CTSD, and CTSL (*Figure 5B*). We propose the following model to explain how estrogen-induced *Ctsb* drives LM-PCD. First, increased gene expression and protein levels of CTSB lead to increased levels of activated CTSB due to heightened lysosomal activity in mammary cells with ongoing LM-PCD. The activated CTSB would further increase the cleavage and activation of CTSB, CTSD, and CTSL. CTSB can also enhance the permeability of lysosomal membrane because lysosomal leakage is markedly decreased in cathepsin B-deficient cells in respond to TNFα treatment (*Werneburg et al., 2002*). Hence, estrogen-induced expression of *Ctsb* triggers a chain of events, culminating in increased LM-PCD. In addition, estrogen-induced expression of *Tnf* and *Bid* further enhances this process. Although estrogen-induced *Tnf* is not associated with increased activation of STAT3, TNFα has been reported to redistribute the zinc transporter ZnT2 to increase Zn in lysosomes. The high levels of Zn cause lysosomal swelling and cathepsin B release during mammary involution (*Hennigar and Kelleher, 2015*; *Hennigar et al., 2015*). Additionally, it is well known that cathepsins cleave BID to tBID and degrade antiapoptotic BCL-2 homologues to execute LM-PCD (*Droga-Mazovec et al., 2008*; *Stoka et al., 2001*). Increased *Bid* level would thus reinforce cathepsin-stimulated LM-PCD. Hence, estrogen-induced increases of cytosolic cathepsins together with increase of *Tnf* and *Bid* accelerate LM-PCD during mammary involution.

On the other hand, estrogen induction of *Ctsb* in nulliparous mice (*Figure 7D*) without ongoing LM-PCD would not lead to cell death due to the lack of a permissive cellular environment. This notion is supported by the study of estrogen in MCF7-Caspase3(+) cells, in which treatment with TNFα primes estrogen to stimulate cell death. In contrast to the observation in involuting mammary gland in which estrogen did not increase total or pSTAT3, estrogen plus TNFα enhanced STAT3 phosphorylation as compared to TNFα alone in MCF7-Caspase3(+) cells (*Figure 6C*). However,

TNFα did not affect the levels of cathepsins significantly in the cell model, and hence it unlikely induced LM-PCD. Nonetheless, the in vitro study demonstrates that estrogen can be primed to heighten cell death under a pro-inflammatory stimulus. Estrogen has also been reported to induce apoptosis in experimental endocrine resistance models through inducing endoplasmic reticulum stress and inflammatory response (*Ariazi et al., 2011*). Thus, estrogen can induce cell death under certain cellular context, but the mechanisms may vary.

Mammary macrophages have also been shown to be important for mammary involution including LM-PCD (*O'Brien et al., 2012*). Depletion of mammary macrophage in Macrophage Fas-induced apoptosis (Mafia) transgenic mice targeting CSF1R-expressing cells resulted in delayed mammary involution and loss of LM-PCD due to attenuated activation of Stat3. Since estrogen induced monocyte recruitment and is known to regulate the transcriptional activity of macrophages (*Chung et al., 2017*; *Pepe et al., 2017*), we speculate that monocytes and macrophages played a part in estrogen-induced LM-PCD by heightening the proinflammatory tissue environment, leading to further activation of estrogen-induced lysosomal proteases.

## Conclusion

In summary, the study reveals novel mechanisms of estrogen action that are unique to the mammary tissue during post-weaning mammary involution and most of the effects are mediated by ERα (*Figure 9*). Estrogen treatment intensely and extensively regulates the activity of neutrophils to elicit inflammation and adipocytes repopulation. The study identified CXCR2 signalling in neutrophils as an important mechanism of estrogen-induced neutrophil recruitment. Additionally, estrogen also exacerbates mammary LM-PCD by inducing the expression of *Ctsb*, *Tnf*, *Bid*, and subsequent lysosomal activation and leakage of CTSB, CTSD, and CTSL independent of neutrophils. At the same time, estrogen retains its function in inducing the expression of pro-growth genes to facilitate mammary regeneration for the subsequent reproduction. It should also be recognized that programmed cell death is often associated with growth, when the elimination of unwanted cells can benefit tissue remodelling and regeneration (*Pérez-Garijo and Steller, 2015*). It has been shown in *Drosophila* eye imaginal discs that apoptotic cells release Spi, the EGF ligand in flies, to promote the proliferation of neighbouring cells (*Fan et al., 2014*). Whether estrogen-induced LM-PCD during mammary involution additionally facilitates mammary regeneration and tumor development is an interesting area for future study.

Both circulating and tissue neutrophils are known to undergo functional and phenotypic changes during cancer development and under other pathological conditions (*Sagiv et al., 2015*; *Nicolás-Ávila et al., 2017*). The observation that neutrophils in the mammary tissue during mammary involution respond to estrogen distinctively from that of nulliparous mice suggests that mammary neutrophils in these two developmental stages are functionally and phenotypically different due to epigenetic differences. Since estrogen promotes mammary tumor growth in mice undergoing mammary involution but not in nulliparous mice (*Chung et al., 2017*), it is tempting to speculate that neutrophils in inflammatory breast cancer may be responsive to the pro-inflammatory effect of estrogen and promote tumor development. Further understanding of the epigenetic changes of mammary neutrophils during involution may yield biomarkers that are predictive pro-tumoral neutrophils.

# Materials and methods

## Animal studies

All animal experiments were performed in accordance with the protocol approved by the Nanyang Technological University Institutional Animal Care and Use Committee (NTU-IACUC) (IACUC protocol number: A0306 and A18036). BALB/cAnNTac mice used in the study were housed in specific pathogen-free (SPF) facility under a 12 hr dark/light cycle and provided food and water ad libitum. Female mice at 7 to 8 weeks old were mated and subsequently housed individually prior to giving birth. Bilateral ovariectomy (OVX) was performed on the lactating mice two days post-partum to remove the endogenous source of estrogen. The litter sizes were standardized to 5–6 pups per lactating mouse. Mammary involution was initiated by forced weaning on lactating day 12. Nulliparous mice were handled similarly to the pregnant mice. All mice were randomly distributed into the different treatment groups.

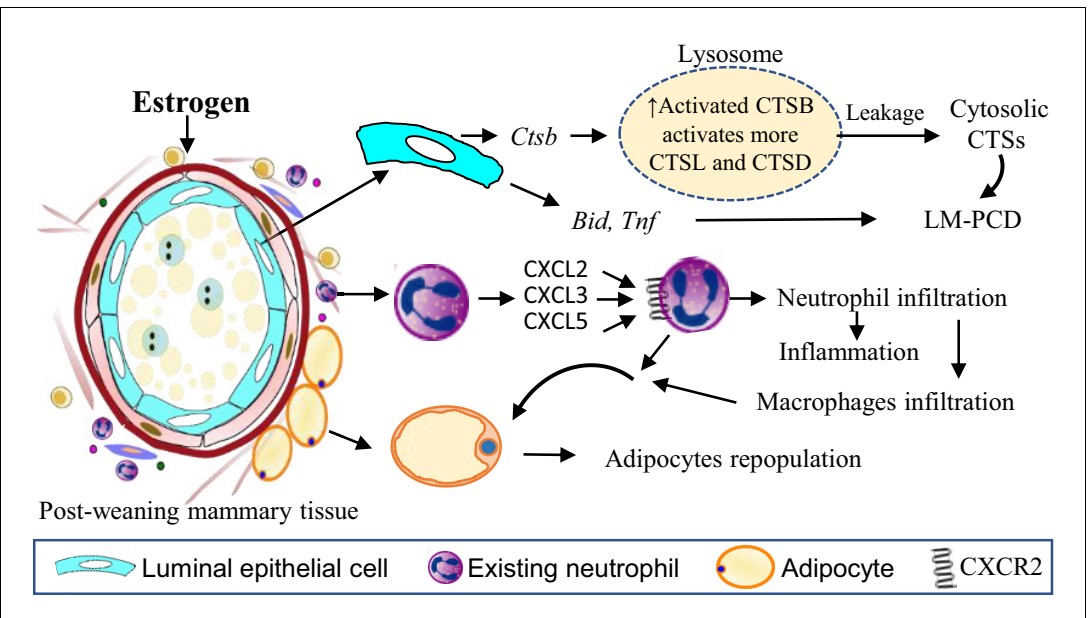

**Figure 9.** Estrogen exacerbates mammary cell death, inflammation and adipocytes repopulation through distinct mechanisms. Estrogen induces *Ctsb* gene expression in the mammary cells leading to increased pro-CTSB which is cleaved and activated in lysosomes. The increased active sc-CTSB further activates CTSD and CTSL. Increased CTSB activity enhanced lysosomal permeabilization during mammary involution resulting in the leakage of more activated CTS into the cytosol stimulating a heightened LM-PCD. Estrogen treatment also induces expression of *Bid* and *Tnf* gene which are reported to be involved in the induction of LM-PCD (*Hennigar and Kelleher, 2015*; *Stoka et al., 2001*). The apoptotic protein BID is cleaved into the active tBID by the activated cytosolic CTSs. TNFα is known to induce LM-PCD via the ZnT2-mediated zinc accumulation in lysosomes, leading to PCD (*Hennigar and Kelleher, 2015*). Estrogen stimulates neutrophil infiltration into the involuting mammary tissue via CXCR2 signalling by up-regulating CXCR2 and its ligands CXCL2, CXCL3, CXCL5 in neutrophils in a autocrine fashion. Meanwhile, estrogen promotes the expression of numerous proinflammatory genes such as *Trem1, Trem3, Il1b, S100a8, S100a9* in neutrophils that likely heighten mammary inflammation. Estrogen induction of genes coding for adipogenic proteins (e.g. CEBPB, Adig and Egr2), for extracellular matrix remodelling enzymes (e.g. Mmp19, Mmp3, Mmp8, Ptx3, Col8a2, Has2) and for Cox-2 etc can facilitate adipocyte repopulation. In addition, estrogen-induced monocytes infiltration and macrophage activity may further contribute to estrogen-induced adipocyte repopulation.

## Drug and estrogen treatment

17β-estradiol-3-benzoate (E2B) (Sigma-Aldrich) dissolved in benzyl alcohol was administered subcutaneously at 20 µg/kg of body weight per day in sesame oil. Control (Ctrl) animals received sesame oil at the corresponding volume per body weight. ER-α-selective agonist propyl pyrazole triol (PPT) (Sigma-Aldrich) was administered subcutaneously at 10 mg/kg of body weight per day in sesame oil. ER-β-selective agonist diarylpropionitrile (DPN) (Sigma-Aldrich) was administered subcutaneously at 5 mg/kg of body weight per day in sesame oil.

CXCR2 inhibitor SB225002 (Sigma-Aldrich) was dissolved in DMSO to a stock concentration of 20 mg/ml. Mice were treated twice daily at a dosage of 0.3 mg/kg body weight via intraperitoneal injection. Stock concentration of SB225002 was diluted to a working concentration of 0.2 mg/ml with 0.25% (v/v) Tween 20 (Bio-Rad) in 1xPhosphate-buffered saline (PBS) (Vivantis). Control mice were treated with DMSO in 0.25% Tween 20 in 1xPBS.

Putative S100A9 inhibitor Paquinimod (PAQ) or ABR-215757 was synthesized (*Figure 3—figure supplement 2*) by following the reported protocol (*Jönsson et al., 2004*). The stock solution was prepared by dissolving PAQ in DMSO to a concentration of 50 mg/ml. Mice were treated with a daily dosage of 20 mg/kg body weight via intraperitoneal injection of PAQ diluted with 1xPBS to a working concentration of 5 mg/ml. Control mice were treated with DMSO in 1xPBS.

### Neutrophil depletion

At 24 hr post-weaning, involution day 1 (INV D1), mice were administered with either rat IgG2a iso-type control (clone 2A3, BioXCell) or rat monoclonal anti-mouse Ly6G antibody (clone 1A8, BioXCell) via intraperitoneal injection. Antibodies were administered at a daily dosage of 20 µg in 100 µl of 1xPBS. For nulliparous animals, antibody treatment was performed at 24 hr prior to hormone injection. The efficiency of neutrophil depletion by the neutralizing antibodies was determined by flow cytometry analysis of the blood and the mammary gland.

### RNA-sequencing (RNA-Seq)

OVX female mice were treated with either anti-Ly6G or IgG antibody at INV D1. At INV D2, mice were then treated with either Ctrl or E2B for 24 hr. Mammary tissues were collected and snap-frozen in liquid nitrogen and stored at $-80°C$. Total RNA was extracted from the powdered $9^{th}$ abdominal mammary gland using TRIzol reagent (Life technologies) followed by treatment with DNase I (DNA-free DNA Removal Kit, Invitrogen) to remove contaminating DNA according to the manufacturers' protocol. The total RNA was then sent for library preparation and paired-end sequencing by A*STAR GIS (Agency for Science, Technology and Research, Genome Institute of Singapore) using Illumina HiSeq4000. The Illumina adapter sequence was removed using Trimmgalore. Processed sequences were subsequently mapped to the *Mus musculus* BALB/cJ reference genome (obtained from Ensembl) and counted using the stringtie and featurecount program. Gene annotation files were also obtained from Ensembl.

Processed RNA-Seq data were analyzed for differential gene expression using the DESeq2 package with contrast method (*Love et al., 2014*). Statistically significant differential gene expression was determined by Benjamini-Hochberg adjusted p-value (padj). Volcano plot based on the results of DESeq2 analysis was generated using the plotly package (*Sievert, 2018*). Venn diagram was plotted using the online software Venny (*Oliveros, 2007*). Pathway analysis of the differentially expressed (DE) gene (padj < 0.05) was then conducted using the clusterProfiler package (*Yu et al., 2012*) where gene ontology (GO) over-representation analysis was performed. The enriched GO terms obtained from the GO over-representation analysis were removed of redundancy using the 'simplify' function which removes highly similar enriched GO terms and keeps only one representative term. The DESeq2, plotly, and clusterProfiler package was run in R using RStudio (*R Studio Team, 2018*; *R Development Core Team, 2019*).

### Isolation of mammary neutrophils

Mammary tissues collected from the euthanized mice were minced into small pieces of approximately 1 mm. Minced mammary tissues were then digested in phenol-red free Dulbecco's Modified Eagle Medium (DMEM) (Nacalai Tesque) containing 2 mM L-glutamine (GE healthcare), 1 mg/ml collagenase (Sigma-Aldrich), and 120 Kunitz DNase I (Sigma-Aldrich) for 1 hr at 37°C in water bath with agitation. Digestion was inactivated with an equal volume of medium and cells were sieved through a 100 µm sieve and centrifuged at 450 g for 5 min at 4°C. Red blood cells (RBC) lysis was then performed with 1 ml $NH_4Cl$ buffer (1 vol of 0.17M Tris-HCL and 9 vol of 0.155M $NH_4Cl$). After inactivation of RBC lysis with 10 vol of 1xPBS, cells were pellet and incubated with 120 Kunitz/ml DNase I for 15 min at room temperature. Cells were then centrifuged again and removed of supernatant before resuspension in isolation buffer (calcium and magnesium-free 1xPBS, 0.1% (w/v) bovine serum albumin (BSA) (Cell Signaling Technology), 2 mM EDTA, pH7.4).

Isolation of mammary neutrophils using Dynabeads (Thermo Fisher Scientific) from the digested mammary cells were carried out following the manufacturer's protocol. Briefly, 5 million cells from the mammary tissue digest were incubated with 1 µg of biotin-anti-Ly6G antibody (clone 1A8, BioLegend) at 4°C for 10 min. After washing, cells were then incubated with 10 µl of Dynabeads for 30 min at 4°C with gentle rotation and tilting. Dynabeads-bound cells were separated from the non-bound cells using a magnetic stand (Milipore) on ice. Isolated Dynabeads bound neutrophils were washed and then added TRIzol, snap-frozen with liquid nitrogen and stored at $-80°C$. A small amount of the non-bound cell fraction after washing was used for flow cytometry analysis of cell depletion efficiency. Remaining non-bound cells were also collected in TRIzol, snap-freeze and stored at $-80°C$. A concurrent separate isolation was also performed with the same above described

protocol with no antibody incubation. This was done to obtain a negative control to ensure the specificity of the Dynabeads isolation process.

## Quantitative PCR (qPCR)

Following the isolation of total RNA with TRIzol, reverse transcription was carried out using qScript cDNA SuperMix (Quantabio) following the manufacturer's protocol. qPCR was carried out with KAPA SYBR FAST qPCR Master Mix (KAPA Biosystems) on the Quantstudio 6 Flex Real-Time PCR System (Applied Biosystems). qPCR for each target gene was performed in duplicates. For quantitative analysis, the comparative Threshold Cycle ($C_t$) method was used, while normalizing to $C_t$ value of *Rplp0* or *Gapdh* in the same sample. Relative quantification was performed using the $2^{-\Delta Ct}$ method (*Schmittgen and Livak, 2008*). The data are expressed as relative mRNA level in arbitrary values. Primers are listed in *Supplementary file 1*.

## Histological analysis

4% paraformaldehyde-fixed mammary tissue samples (4th abdominal mammary gland) were paraffin-embedded and sectioned at 5 µm for haematoxylin and eosin (H and E) staining and immunohistochemical (IHC) staining. To quantify epithelial cell death, the number of dying cells with hyper-condensed nuclei shed into the alveolar lumen of the mammary gland was counted. Cleaved caspase3+ cells were identified by immunostaining with antibody against cleaved caspase 3. IHC staining was carried out using the VECTASTAIN Elite ABC Kit (Vector laboratories) and perilipin A (1:100, D418 Cell Signaling) or cleaved caspase 3 (1:100, #9661 Cell Signaling) followed by the DAB (3,3'-diaminobenzidine) peroxidase (HRP) substrate kit (with Nickel) (Vector Laboratories). The tissue sections were counterstained with Richard-Allan Scientific Signature Series Hematoxylin 2. All histological analysis was performed with images from at least five random fields for each sample.

## Subcellular fractionation

Subcellular fractionation of mammary glands was carried out based on the published protocol (*Kreuzaler et al., 2011*). Briefly, mammary tissues in liquid nitrogen were powdered and homogenized in a handheld homogenizer in subcellular fractionation buffer (20 mM HEPES- KOH, 250 mM sucrose, 10 mM KCl, 1.5 mM $MgCl_2$, 1 mM EDTA, 1 mM EGTA, 8 mM dithiothreitol, 1 mM Pefabloc, at pH7.5) and centrifuged at 750 g for 10 min at 4℃ to remove cell nuclei and debris. The supernatant was then spun at 10,000 g for 15 min at 4℃ to pellet organelles. The pellet was washed and re-suspended in subcellular fractionation buffer as lysosomal fraction. Organelles were disrupted by three cycles of freezing and thawing. To collect the cytosolic fraction, the supernatant collected after pelleting organelles was spun at 100,000 g for 1 hr at 4℃ to remove microsomes. Protein concentration was determined by the Bradford protein assay (Bio-rad).

## Flow cytometry analysis

Blood cells collected via cardiac puncture were lysed of RBC via incubation with $NH_4Cl$ buffer for 15 min at room temperature with gentle tilting. RBC lysis was inactivated with equal volume of 1xPBS followed by centrifugation at 450 g for 5 min at 4℃ and resuspended in 1xPBS. Mammary tissues were digested as previously described until after the RBC lysis step in which mammary cells were subsequently reconstituted in 1xPBS. Staining of cells was carried out with a cocktail of primary antibodies from either BioLegend or eBioscience comprising of APC-Cy7-anti-CD45 (clone 30-F11), FITC-anti-Ly6C (clone HK1.4), PE-anti-Ly6G (clone 1A8), Biotin-anti-Gr1 (clone RB6-8C5), and BV605-anti-CD11b (clone M1/70). Secondary antibody staining was performed with Alexa Fluor 647-streptavidin antibody (BioLegend). Dead cells were stained with eFluor 450-fixable viability dye (eBioscience) before fixation with fixative buffer (BioLegend).

## Protein collection and western blotting

Proteins were isolated from cells or tissues by adding cold lysis buffer containing 50 mM HEPES (pH7.5), 150 mM NaCl, 100 mM NaF, 1 mM PMSF, 1% (v/v) Triton X-100, and a cocktail of proteinase inhibitors (2 ug/ml aprotinin, 5 ug/ml leupeptin, 1 mM Na3VO4, and 5 ug/ml pepstatin A). After lysis, cells were centrifuged at 13800 rpm for 15 min at 4℃ and the supernatant collected. Protein concentration was determined with the BCA protein assay kit (Thermo Fisher Scientific) following the

manufacturer's protocol. The collected protein lysate supernatant was added 5x Laemmli sample buffer and stored at −80℃. Protein samples collected were analyzed with western blotting. The protein band of interest was subsequently quantitated (normalized to the reference protein band) using quantity one software (Bio-rad). Antibodies used for western blot analysis are listed in *Supplementary file 2*.

### In vitro studies of estrogen-induced cell death

MCF7-caspase3(+) cells originally from Porter AG (*Jänicke et al., 1998*) were used in order to observe TNFα-induced apoptosis. The cells were not authenticated since it is only for a very minor portion of the study. However, the cells have been tested negative for mycoplasma using PCR analysis prior to usage. Cells were maintained in DMEM containing 2 mM L-glutamine and 7.5% fetal calf serum (FCS) (HyClone) and kept at 37℃ in a humidified 5% carbon dioxide and 95% air atmosphere.

For estrogen treatment, phenol-red free DMEM supplemented with 2 mM L-glutamine and 5% DCC-FCS (fetal calf serum treated with dextran-coated charcoal) was used. Treatment of FCS with dextran-coated charcoal was performed to remove steroid hormones present in the FCS. MCF7-caspase3(+) cells were plated at 150 k in 6-well plate (Corning) with the DCC-FCS supplemented medium. After 48 hr, the medium was replaced, and cells were treated with either TNFα (ProSpec) diluted in 1xPBS or vehicle control. One hour after TNFα treatment, cells were then treated with 10 nM 17β-estradiol (E2) diluted in 100% ethanol or vehicle control. MCF7-caspase3(+) cells were harvested for analysis 24 hr after E2 treatment. Both floating dead cells and the adherent live cells were collected. Harvested cells were resuspended in 1xPBS and stained with 0.1 µg PI per 100 ul of 1xPBS for 15 min at room temperature. After staining, 400 µl of 1xPBS were added and cells were immediately analyzed with the flow cytometer.

### Statistical analysis

Graphs were plotted using the mean value with the standard error of the mean (SEM). When comparing two groups, statistical significance was determined using a two-tailed unpaired student's t-test. When comparing between more than two groups, one-way ANOVA followed by post-hoc turkey test was performed. All statistical analysis was performed using the GraphPad Prism seven software. p-value:$<0.05$ (*),$<0.01$ (**),$<0.001$ (***),$<0.0001$ (****).

## Acknowledgements

This research is funded the Ministry of Education of Singapore. Academic Research Fund Tier I, MOE2017-T1-002-081. We thank Drs. Natasa Bajalovic, Amanda Woo and Mr. Lee Shi Hao for their technical assistance.

## Additional information

### Funding

| Funder | Grant reference number | Author |
|---|---|---|
| Ministry of Education of Singapore | MOE2017-T1-002-081 | Valerie Chun Ling Lin |

The funders had no role in study design, data collection and interpretation, or the decision to submit the work for publication.

### Author contributions

Chew Leng Lim, Yu Zuan Or, Data curation, Formal analysis, Validation, Investigation, Visualization, Writing - original draft, Writing - review and editing; Zoe Ong, Hwa Hwa Chung, Formal analysis, Investigation, Writing - review and editing; Hirohito Hayashi, Shunsuke Chiba, Resources; Smeeta Shrestha, Formal analysis, Writing - review and editing; Feng Lin, Supervision; Valerie Chun Ling Lin, Conceptualization, Resources, Formal analysis, Supervision, Funding acquisition, Validation, Visualization, Project administration, Writing - review and editing

Author ORCIDs

Chew Leng Lim  https://orcid.org/0000-0003-4529-2732
Smeeta Shrestha  http://orcid.org/0000-0002-6560-4230
Valerie Chun Ling Lin  https://orcid.org/0000-0002-7997-2771

Ethics

Animal experimentation: All animal experiments were performed in accordance with the protocol approved by the Nanyang Technological University Institutional Animal Care and Use Committee (NTU-IACUC) under the protocol number A0306 and A18036.

Decision letter and Author response

Decision letter https://doi.org/10.7554/eLife.57274.sa1
Author response https://doi.org/10.7554/eLife.57274.sa2

# Additional files

## Supplementary files

- Supplementary file 1. List of qPCR primers used.
- Supplementary file 2. List of antibodies used.
- Transparent reporting form

## Data availability

Sequencing data have been deposited in DR-NTU (DATA) accessible with the URL https://doi.org/10.21979/N9/YBRINN.

The following dataset was generated:

| Author(s) | Year | Dataset title | Dataset URL | Database and Identifier |
|---|---|---|---|---|
| Lim CL | 2020 | RNA-sequencing data of Balb/c involuting mammary gland treated with anti-Ly6G antibody and estrogen | https://doi.org/10.21979/N9/YBRINN | DR-NTU (DATA), 10.21979/N9/YBRINN |

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
