## [Decision Letter]

**Acceptance summary:**

The microenvironment of the involuting mammary gland is an important determinant of pregnancy- and lactation-associated breast cancer. This study revealed interesting links between a female sex hormone and mammary involution, including estrogen-triggered neutrophil accumulation and CXCR2 signaling that exacerbated local inflammation, as well as lysosome-mediated programmed death of mammary cells.

**Decision letter after peer review:**

Thank you for submitting your article "Estrogen exacerbates mammary involution through neutrophil dependent and independent mechanism" for consideration by *eLife*. Your article has been reviewed by three peer reviewers, one of whom is a member of our Board of Reviewing Editors, and the evaluation has been overseen by Tadatsugu Taniguchi as the Senior Editor. The reviewers have opted to remain anonymous.

The reviewers have discussed the reviews with one another and the Reviewing Editor has drafted this decision to help you prepare a revised submission.

Summary:

Lin and colleagues aim to explore how estrogen promotes post-partum mammary involution and increases the risk of parity-associated breast cancer. Previous studies have unraveled different molecular mechanisms of mammary gland involution (see a previous summary PMID: 30448440). The authors highlighted that estrogen causes mammary involution by stimulating the accumulation of neutrophils, sculpturing the pro-inflammatory microenvironment, facilitating adipocytes differentiation, and causing lysosome-related programmed death of mammary cells. These findings are interesting, but further efforts are needed to nail down some of the major conclusions and to clarify the underlying mechanisms.

Essential revisions:

1) A major concern is about several discoveries on neutrophils. The contribution of neutrophils during estrogen-induced mammary involution should be cautiously defined with solid experimental evidence. Do other immune cell populations, such as macrophages, actively participate in this process? Does Ly6G antibody efficiently deplete neutrophils, rather than masking the labeling of Gr1 antibody (for validation)? As neutrophils have short half-life, secret lots of inflammatory mediators, and quickly replenish from BM progenitors, this point is important. The possible coordinations between neutrophils and other immune cells (e.g. macrophages, monocytes), and their relative importance at different stages of mammary involution can be examined and discussed (see a previous summary PMID: 24952477). In addition, more evidences are needed to prove whether the recruitment of neutrophils depends on a positive-feedback loop of CXCL1 and CXCL2. Quality controls are needed for the application of inhibitors.

2) Another major concern is about estrogen-triggered mammary cell death. How do ERα, ERβ or death receptor-mediated signals contribute to this process? Does estrogen-induced programmed cell death exclusively rely on lysosome leakage and related effector molecules? Have the authors tested the existence of other cell death modalities? Does estrogen-induced cell death augment local inflammation and perhaps the accumulation of immune cell populations?

3) The authors may consider to rephrase/weaken some of their claims and reorder the display of some of their results.

Reviewer #1:

Dr. Valerie Lin discovered some interesting links between estrogen signals and the differentiation and programmed death of mammary cells, as well as the formation of pro-inflammatory microenvironment, which facilitate post-partum mammary involution and presumably parity-associated breast cancer. They also demonstrated that mammary gland-infiltrating neutrophils emerge as a major immune cell participant during this process.

1) Dr Shengtao Zhou reported that ERβ has potent antitumor effects, which suppress lung metastasis by recruiting antitumor neutrophils to the metastatic niche. It is recommended to carefully address whether estrogen specifically target ERα or ERβ on post-weaning mammary cells and infiltrating neutrophils in this setting.

2) Perhaps a missing point is whether estrogen-induced mammary cell death subsequently cause inflammation (presumably augmenting neutrophil accumulation), since several cell death modalities has been associated to inflammation via the release of danger molecules.

3) Besides lysosome-mediated PCD, can estrogen induce other cell death modalities, such as pyroptosis, necroptosis, ferroptosis, which are all caspase 3/7/8-independent? It is important to make a clear conclusion.

4) Since the synthetic estrogen regulate the differentiation associated genes of fat cells and accelerated LM-PCD. Does estrogen affect lipid metabolic pathways? How does this metabolic remodeling affect cell death and differentiation of adipocytes and the function of neutrophils? By carefully going over the Seq data, the authors may add more important discussions.

Reviewer #2:

In this study, Chew Leng Lim et al. determined the diverse effects of Estrogen exposure on neutrophil infiltration, inflammation responses, cell death and adipocytes repopulation in mice models. While the authors revealed some new findings, this study suffers from obvious defects, including overdependence on the use of chemical inhibitors, lack of in-depth mechanistic investigation as well as unfocused research topics.

1) In addition to neutrophil, estrogen exposure also induced macrophage infiltration, while "neutrophil" deletion by using anti-Ly6G antibody obviously reduced the infiltration of macrophage (Figure 2—figure supplement 1). Therefore, the role of macrophage in Estrogen exposure-induced biological responses should be deeply determined.

2) Since anti-Ly6G antibody also reduced macrophage infiltration significantly, it is very likely macrophage play a pivotal role in Estrogen exposure-regulated gene expressions and cellular phenotypes. Therefore, the conclusion that 88% of estrogen-regulated genes are mediated through neutrophil is not solid. This point should be addressed by specific deletion of macrophage and neutrophil, individually.

3) The source of CXCL1/CXCL2 upon estrogen exposure should be further investigated. In Figure 8, the authors indicated that CXCL1 and CXCL2 are produced by existing neutrophil. Further evidence to support this should be provided.

4) Many biological and small molecule inhibitors, including anti-Ly6G antibody, PAQ (S100a9 inhibitor), CXCR2 antagonist SB225002, etc, have been frequently used in this study. However, the effects and specificity of some of these agents have not been well validated during the study. The use of genetic mice models for critical signaling pathways is highly suggested.

5) Figure 5, The critical role of CTSB in the activation of CTSD/CTSL and induction of LM-PCD upon E2B treatment should be validated by downregulation of CTSB expression pharmacologically or genetically.

6) The data presented in Figure 7 based on the analysis in a single cell line is not reliable.

Reviewer #3:

This paper reported that estrogen can accelerate mammary involution by exacerbating mammary inflammation, inducing programmed cell death, and promoting adipocytes repopulation, that the effects of estrogen on the expressions of genes during mammary involution are majorly mediated by neutrophils, and that estrogen promotes mammary LM-PCD independent of neutrophils by inducing the expression and activity of lysosomal cathepsins and other pro-apoptotic markers such as *Bid* and *Tnf*. These findings are potentially interesting, and could expand the functions of estrogen. However, there is a lack in mechanistic insight into these observations. I. The mechanism underlying estrogen-induced cell death needs to be further explored. For example, what kind of player(s) connects estrogen with cell death? Whether TNF-α plays a role in linking estrogen to cell death? Is there any enrichment of cell death genes associated with the estrogen treatment in RNA-Seq data? Why the artificial MCF-7/Caspase-3 cells were used? The results about MCF-7/Caspase-3 cells showed that estrogen promoted TNF-α-induced apoptosis, rather than lysosomes-associated cell death. Maybe the authors should try MCF-10A cells as the model. II. Based on the data on Figure 4, it is not so convincible to conclude that neutrophils are involved in adipocytes repopulation during mammary involution normally, please see also Issues#3. The authors need to re-consider the relationship between these data and the conclusion. Maybe they should re-describe these results or modify the conclusion. III. The most interesting finding is that estrogen does not trigger the similar biological actions in age-matched nulliparous mammary tissue. However, this study does not figure out the molecular mechanism underneath the difference between the functions of estrogen in involutional and nulliparous mammary tissues. At least, the author should discuss about the potential possibilities.

Other issues:

1) Quantification in Figure 1B should indicate the fractions, for example, No. cells of total or area. The data in Figure 1C, except Csn2, were not described in the content, and these data should be associated with adipogenesis. As for Figure 1D, no any description was presented about Ly6G, and in fact, it was described in the second part of Results section. Supplemental Figure 2 was mentioned in the content before Supplemental Figure 1. The first part of results was very important for readers to understand the paper, but these problems confuse the readers.

2) "E2B treatment alone without the antagonist (E2B+DMSO) lead to an expected 1.57-fold increase (p=0.0082) in mammary neutrophils as compared to the Ctrl+DMSO". Should "1.57-fold" be "2.57-fold" or something other? It is not the case based on the data in Figure 3Ci.

3) In Figure 4B, upon neutrophil depletion, *Cebpb* and *Cebpd* were already increased, which could limit their further enhancement when treated with E2B. As for Adig and Egr2, it seemed that they also apparently increased. In Figure 4D, the data had the similar problem to those in Figure 4B. No description about Figure 4E and 4F was found in the content. Overall, these data put it in question that estrogen-induced adipocyte repopulation is associated with the induction of adipogenic and tissue remodeling genes through neutrophils.

4) "This suggests that the up-regulation of *Ctsb* expression by E2B is a direct event independent of STAT3 activation". These data in Figure 5B could not demonstrate that *Ctsb* expression is the direct event of E2B. In Figure 5D, why the lysosomal pellet fractions showed no lysosomal proteins, such as catheptins. In Figure 5A, at least the protein level of TNF-α should be measured, because it was very important for the functions of E2B, based on the data in Figure 6.

5) In Figure 6, TNF induces p-STAT3 while Figure 5A shows E2 induces TNF expression (mRNA), but no p-STAT3 was increased in Figure 5B. The increased mRNA does not mean the increased protein. Please measure the TNFα proteins in Figure 5A (See Issue#4). The MCF-7/Casp3 model seems not to well support the conclusion. The data in Figure 6 are about typical apoptosis not the lysosomes-associated cell death involved in the functions of estrogen as revealed in this study.

---

## [Author Response]

Essential revisions:1) A major concern is about several discoveries on neutrophils. The contribution of neutrophils during estrogen-induced mammary involution should be cautiously defined with solid experimental evidence. Do other immune cell populations, such as macrophages, actively participate in this process?

We agree that mammary monocytes and macrophages may also contribute to estrogen-induced mammary involution. During sterile inflammation, infiltrated neutrophils was known to recruit macrophages into the tissue from the circulation [1]. We also reported that estrogen induced recruitment of monocytes [2] (Supplementary Figure 2), although the functional consequence of this recruitment has not been looked at. Additionally, neutrophils depletion reduced mammary monocytes, albeit not to a statistically significant degree. Macrophages have been reported to be critical for cell death and adipocyte repopulation during mammary involution [3]. Whilst we did not investigate the role of estrogen-induced monocytes during mammary involution, estrogen is also known to regulate gene activity in peritoneal macrophages [4].

The possible involvement of mammary monocytes/macrophages in estrogen-induced adipocyte repopulation and LM-PCD have been discussed in the revised manuscript.

In Supplementary Figure 2 of the previous study estrogen significantly induced monocytes infiltration, but not F4/80 macrophages. Although it marginally (p=0.0538 for CD4 cells) reduced the percentage of CD4 and CD8 cells, this decrease is likely due to increases in the percentage of neutrophils and monocytes.

Does Ly6G antibody efficiently deplete neutrophils, rather than masking the labeling of Gr1 antibody (for validation)? As neutrophils have short half-life, secret lots of inflammatory mediators, and quickly replenish from BM progenitors, this point is important.

Anti-Ly6G antibody (clone 1A8) is widely known to effectively and specifically deplete neutrophils. Furthermore, neutrophils has been demonstrated to co-stain with both clone 1A8 anti-Ly6G and clone RB6-8C5 anti-Gr1 [5], suggesting that the two antibodies target different epitopes. In our hands through a number of experiments, Ly6G antibody consistently reduced blood and mammary neutrophils by more than 90%. Therefore, the use of Gr1 antibody clone RB6-8C5 to check for neutrophil depletion efficiency by Ly6G antibody (clone 1A8) is valid and reliable.

The possible coordinations between neutrophils and other immune cells (e.g. macrophages, monocytes), and their relative importance at different stages of mammary involution can be examined and discussed (see a previous summary PMID: 24952477).

This has been addressed above.

In addition, more evidences are needed to prove whether the recruitment of neutrophils depends on a positive-feedback loop of CXCL1 and CXCL2. Quality controls are needed for the application of inhibitors.

The involvement of CXCR2 signalling in neutrophil recruitment to tissues has been widely reported [6-9]. CXCR2 receives input from several chemokine ligand. In mice, CXCR2 ligands include CXCL1, CXCL2, CXCL3 and CXCL5 [10]. Our RNA-Seq analysis showed that estrogen up-regulated the expression of all four chemokines and neutrophil depletion abolished their upregulation (Figure 3A) Furthermore, we provide new data in Figure 3B that the upregulation of Cxcl2, Cxcl3, and Cxcl5 occurs in isolated neutrophils by qPCR analysis. We propose that the recruitment of neutrophils involves an autocrine mechanism, in which estrogen induces neutrophil release of chemokines which in turn recruit neutrophils through CXCR2 signalling.

We have revised the manuscript and according to this analysis. We also revised the model incorporating this new data.

As for the comment on a positive-feedback loop for estrogen-induced neutrophil recruitment, we agree that it is a logical and attractive deduction of our current data. Positive feedback refers to the mechanism in which a change triggers a response that enhances the change. It is logical to deduce that estrogen-induced upregulation of Cxcls in neutrophils (change) that causes neutrophil recruitment (response) would enhance Cxcl2 release (change) as more neutrophils are recruited. However, this would need a time course experiment to demonstrate an incremental increase of mammary neutrophils recruitment in response to estrogen overtime. This time course experiment would require a large number of mice (3 time points x 4 mice per treatment x 4 groups, at a minimum) and approval from institutional animal care and use committee. We would like to seek your understanding that we are unable to justify for this experiment to test a hypothesis that seems to be a logical deduction.

2) Another major concern is about estrogen-triggered mammary cell death. How do ERα, ERβ or death receptor-mediated signals contribute to this process?

To elucidate whether ERα or ERβ mediates the effect of E2B on various process of mammary involution, we evaluated the effect of ERα-specific agonist, (4,4',4''-(4-Propyl-[1H]-pyrazole-1,3,5-triyl) trisphenol (PPT) [11] and ERβ-specific agonist Diarylpropionitrile (DPN) [12]. Similar to the effect of E2B, ERα agonist PPT evidently increased the expression *Ctsb*, *Bid* and *Tnf* that are known to promote cell death (Figure 8A). PPT also consistently increased the protein levels of cleaved CTSB (sc-CTSB), cleaved CTSD (sc-CTSD), and cleaved CTSL (sc-CTSL and dc-CTSL) (Figure 8B), which are executioner proteases [13, 14]. In contrast, ERβ-specific agonist DPN did not have any effect on the gene expression of the death-inducing genes, or the activation of lysosomal proteases. These observations indicate that ERα mediates E2B-induced LM-PCD during mammary involution.

The results of PPT and DPN are described under the section “Estrogenic stimulation of mammary inflammation and mammary cell death is mediated by ERα”.

Does estrogen-induced programmed cell death exclusively rely on lysosome leakage and related effector molecules? Have the authors tested the existence of other cell death modalities?

Estrogen-induced mammary cell death was evaluated at 48h post-weaning. Mammary involution in the first 48h is known as phase I (reversible phase) of mammary involution. In phase I, LM-PCD was found to be the major mechanism of mammary cell death [15]. The evidence obtained in this study also indicates that estrogen induces LM-PCD by inducing the expression of *Ctsb*, *TNF* and *Bid*, and the activation of CTSB, D and L.

Although we have not tested experimentally if estrogen induced other forms of cell death, we did look through the RNA-Seq data related to cell death and survival. We could not find any consistent pattern to suggest additional mechanism of estrogen-induced cell death. For example, we found that estrogen induced the expression of CRADD which is known to recruit Casp2 to promote cell apoptosis [16]. However, estrogen down-regulated Casp2 independent of neutrophils, which is inconsistent with the involvement of CRADD-Casp2 pathway in estrogen-induced cell death (Figure 5—figure supplement 1).

We also noticed that estrogen induces Ripk3 expression that is dependent on neutrophils (Figure 5—figure supplement 1), which we subsequently validated by qPCR. Ripk3 is an important modulator of necroptosis in which cells release reactive molecules like DAMPs (damage-associated molecular patterns) into the tissue triggering inflammation [17-19]. RIPK3 activation is important in the formation of neutrophil extracellular traps and the subsequent neutrophil death in venous thrombosis model [20]. It can be hypothesized that estrogen-induced upregulation of RIPK3 in neutrophils affects neutrophils survival, but this would require further evaluation.

We have added the following in the revised manuscript to lend support to the idea that estrogen-induced cells death is mainly mediated through LM-PCD: “Mammary involution in the first 48h is known as phase I (reversible phase) of mammary involution. In phase I, LM-PCD has been shown to be the major mechanism of mammary cell death, and it is independent of caspases 3, 6 and 7 [15]. [...] Hence, estrogen-induced increases of cytosolic cathepsins together with increase of *Tnfα* and *Bid* accelerate LM-PCD during mammary involution.”

Does estrogen-induced cell death augment local inflammation and perhaps the accumulation of immune cell populations?

We have added the following in the Discussion:

“It is well known that dying cells release signalling molecules to stimulate inflammation and recruitment of phagocytic cells for the clearance of dying cells [21]. Likewise, estrogen-induced cell death may aggravates inflammation and immune cell recruitment. This may further contribute to neutrophil recruitment in response to estrogen in mammary involution model.”

We also added additional discussion under the section “Estrogen promotes LM-PCD during mammary involution”

“Mammary macrophages have also been shown to be important for mammary involution including LM-PCD [3]. Depletion of mammary macrophage in Macrophage Fas-induced apoptosis (Mafia) transgenic mice targeting CSF1R-expressing cells resulted in delayed mammary involution and loss of LM-PCD due to attenuated activation of Stat3. Since estrogen induced monocyte recruitment and is reported to regulate the transcriptional activity of macrophages [2, 4], we speculate that monocytes and macrophages played a part in estrogen-induced LM-PCD by heightening the proinflamamtory tissue environment, leading to further activation of estrogen-induced lysosomal proteases”.

3) The authors may consider to rephrase/weaken some of their claims and reorder the display of some of their results.

We have gone through the manuscript to rephrase some of the writing.

Reviewer #1:1) Dr Shengtao Zhou reported that ERβ has potent antitumor effects, which suppress lung metastasis by recruiting antitumor neutrophils to the metastatic niche. It is recommended to carefully address whether estrogen specifically target ERα or ERβ on post-weaning mammary cells and infiltrating neutrophils in this setting.

We added the following to the discussion “It should be recognized that ERβ has been reported to play a role in neutrophil regulation of tumour biology. Selective ERβ agonist LY500307 was found to reduce lung metastasis of triple-negative breast cancer cells [22]. This was associated with significant increase of infiltration of neutrophils in the lung. However, this was mediated by the up-regulation of IL-1β in cancer cells which promotes neutrophil infiltration to the metastatic niche. There was no evidence whether ERβ agonist regulated the phenotypes of neutrophils directly.”

2) Perhaps a missing point is whether estrogen-induced mammary cell death subsequently cause inflammation (presumably augmenting neutrophil accumulation), since several cell death modalities has been associated to inflammation via the release of danger molecules.

This point has been addressed in revised manuscript.

3) Besides lysosome-mediated PCD, can estrogen induce other cell death modalities, such as pyroptosis, necroptosis, ferroptosis, which are all caspase 3/7/8-independent? It is important to make a clear conclusion.

This is addressed in revised manuscript.

4) Since the synthetic estrogen regulate the differentiation associated genes of fat cells and accelerated LM-PCD. Does estrogen affect lipid metabolic pathways? How does this metabolic remodeling affect cell death and differentiation of adipocytes and the function of neutrophils? By carefully going over the Seq data, the authors may add more important discussions.

GO analysis revealed that estrogen regulated genes in the category of "Regulation of lipid metabolic process”. 67 genes are grouped in this category. We described and discussed the significance of top 5 most regulated genes in this category based on fold changes. A general impression is that the gene regulation by estrogen points to both pro- and anti-adipogenic effect.

Reviewer #2:1) In addition to neutrophil, estrogen exposure also induced macrophage infiltration, while "neutrophil" deletion by using anti-Ly6G antibody obviously reduced the infiltration of macrophage (Figure 2—figure supplement 1). Therefore, the role of macrophage in Estrogen exposure-induced biological responses should be deeply determined.

We agree that estrogen-induced increase in mammary monocytes (macrophages) may contribute to estrogen-induced involution. This point has been addressed above and in the revised manuscript.

2) Since anti-Ly6G antibody also reduced macrophage infiltration significantly, it is very likely macrophage play a pivotal role in Estrogen exposure-regulated gene expressions and cellular phenotypes. Therefore, the conclusion that 88% of estrogen-regulated genes are mediated through neutrophil is not solid. This point should be addressed by specific deletion of macrophage and neutrophil, individually.

Anti-Ly6G antibody reduced monocytes infiltration marginally (p>0.05). Nonetheless, we also recognize the potential involvement of macrophages and infiltrating monocytes in estrogen-regulation of mammary involution and discuss as above. Our conclusion is “This 88% of the genes was regulated by estrogen either in neutrophils, or regulated as a result of neutrophil activities on other cells”, which include neutrophil-recruited immune cells such as monocytes. The effect of estrogen on the activity of macrophages and infiltrated monocytes was not investigated in this study.

3) The source of CXCL1/CXCL2 upon estrogen exposure should be further investigated. In Figure 8, the authors indicated that CXCL1 and CXCL2 are produced by existing neutrophil. Further evidence to support this should be provided.

This point has been addressed in the early part of the letter and in the manuscript.

4) Many biological and small molecule inhibitors, including anti-Ly6G antibody, PAQ (S100a9 inhibitor), CXCR2 antagonist SB225002, etc, have been frequently used in this study. However, the effects and specificity of some of these agents have not been well validated during the study. The use of genetic mice models for critical signaling pathways is highly suggested.

The specificity and efficacy of CXCR2 inhibitor SB225002 and the anti-Ly6G antibody clone 1A8 have been well-validated by other studies. Our data using these 2 reagents are consistent with those reported in the literature.

Paquinimod is a derivative of quinoline-3-carboxamide that has been shown to inhibit S100A9 activity through binding with S100A9 [23, 24]. The binding inhibits S100A9 dimerization or the formation of heterodimer with S100A8, thereby preventing the activation of RAGE or TLR4. However, reports on Paquinimod as an inhibitor of S100A9 in the context of neutrophils infiltration is not very well defined. In a study looking at the effect of Paquinimod on immune cell infiltration during sterile peritoneal inflammation, Paquinimod reduced necrotic tumour cell-induced accumulation of monocytes and eosinophils, but not that of neutrophils [25]. Other study showed effectiveness of Paquinimod in inhibiting neutrophils-derived S100A8/A9-mediated inflammatory response, but not specifically on neutrophil infiltration [26, 27]. Contrary to our expectation, Paquinimod increased neutrophil infiltration and the expression levels of S100A8 and S100A9 in estrogen-treated mice (Figure 3—figure supplement 1). This suggests that the functional properties of Paquinimod have yet to be well characterized. The specificity of paquinimod during mammary involution require more evaluation. Bu we do not have the resources to conduct further experiments to understand why it behaved unexpectedly. We thought to include the data to benefit readers who may be interested in this unusual property of Paquiminod.

While we acknowledge the importance of validating the effect of S100A9 with genetic mice model, we did not have access to these mice to embark in these experiments for the current study. We moved on to characterize the involvement of CXCR2 using its chemical inhibitor SB225002.

5) Figure 5, The critical role of CTSB in the activation of CTSD/CTSL and induction of LM-PCD upon E2B treatment should be validated by downregulation of CTSB expression pharmacologically or genetically.We thought about testing CTSB inhibitor CA-074ME but decided against it because it has already been shown to inhibit LM-PCD during mammary involution [15]. We believe that CTSB inhibitor will inhibit LM-PCD in our estrogen–treated mouse model, but it would not tell us whether it inhibits estrogen-induced LM-PCD or just the LM-PCD in general because they are the same process.6) The data presented in Figure 7 based on the analysis in a single cell line is not reliable.Since the observation of estrogen-induced cell death is novel under physiological condition, the purpose of the experiment in Figure 6 in MCF7-caspase3(+) cells was to show that estrogen can induce cell death when primed by certain proinflammatory factors such as TNFα. The results was reproducible from 3 independent experiments and hence proved that estrogen under certain pro-inflammatory condition can induce cell death.Reviewer #3:I) The mechanism underlying estrogen-induced cell death needs to be further explored. For example, what kind of player(s) connects estrogen with cell death? Whether TNF-α plays a role in linking estrogen to cell death? Is there any enrichment of cell death genes associated with the estrogen treatment in RNA-Seq data? Why the artificial MCF-7/Caspase-3 cells were used? The results about MCF-7/Caspase-3 cells showed that estrogen promoted TNF-α-induced apoptosis, rather than lysosomes-associated cell death. Maybe the authors should try MCF-10A cells as the model.

The mechanism of estrogen-induced cell death is explained as the following:

“We propose the following model to explain how estrogen-induced *Ctsb* drives LM-PCD. First, increased gene expression and protein levels of CTSB lead to increased levels of activated CTSB due to heightened lysosomal activity in mammary cells with ongoing LM-PCD. […] Hence, estrogen-induced increases of cytosolic cathepsins together with increase of *Tnfα* and *Bid* accelerate LM-PCD during mammary involution.”

Regarding enrichment of cell death genes associated with the estrogen treatment in RNA-Seq data, the heat map of the list of genes associated with cell death is included in Figure 5—figure supplement 1. We did further analysis on *Tnfα*, *Bid* and *Ctsb* as is shown in Figure 5.

“why the artificial MCF-7/Caspase-3 cells were used? The results about MCF-7/Caspase-3 cells showed that estrogen promoted TNF-α-induced apoptosis, rather than lysosomes-associated cell death. Maybe the authors should try MCF-10A cells as the model.”

Since this is the first evidence that estrogen as a well-known mitogenic hormone elicits cell death under physiological condition, we wanted to show that estrogen can heighten cell death when primed by proinflammatory cytokines such as TNFα. The parental MCF-7 cell line is caspase 3-negative and is not responsive to some apoptotic stimuli [33]. Caspase 3 expression sensitizes MCF-7 cells to apoptosis inducers [34], and is commonly used for studying cell death-related mechanisms. To ensure the model we use are sensitive to TNFα-induced cell death, we choose to use MCF7-caspase3(+). The result proved our hypothesis, but the mechanism of cell death in MCF7-caspase3(+) cells is not LM-PCD.

MCF-10A does not express estrogen receptor and hence is not suitable for the evaluation of estrogen’s effect.

II) Based on the data on Figure 4, it is not so convincible to conclude that neutrophils are involved in adipocytes repopulation during mammary involution normally, please see also Issues#3. The authors need to re-consider the relationship between these data and the conclusion. Maybe they should re-describe these results or modify the conclusion.

We agree that macrophages and monocytes may also be involved. We have addressed this point in the revised manuscript.

III) The most interesting finding is that estrogen does not trigger the similar biological actions in age-matched nulliparous mammary tissue. However, this study does not figure out the molecular mechanism underneath the difference between the functions of estrogen in involutional and nulliparous mammary tissues. At least, the author should discuss about the potential possibilities.

We agree with the reviewer’s suggestion. Discussion of possible mechanisms and significance of estrogen’s plasticity on neutrophils is added in the revised manuscript.

Other issues:1) Quantification in Figure 1B should indicate the fractions, for example, No. cells of total or area. The data in Figure 1C, except Csn2, were not described in the content, and these data should be associated with adipogenesis. As for Figure 1D, no any description was presented about Ly6G, and in fact, it was described in the second part of Results section. Supplemental Figure 2 was mentioned in the content before Supplemental Figure 1. The first part of results was very important for readers to understand the paper, but these problems confuse the readers.

Thank you for pointing them out.

In Figure 1B, the number of shed cells is the total of 5 random field per mouse. For CC3^+^ cells, it is the average of CC3^+^ cells of 5 random field per mice. The reason to present as total number of shed cells from 5 random fields is that it is harder to discern the shed cells with hyper-condensed nuclei and the number is small and more variable among different field. We though it is better to use the sum instead of the average. In contrast, identification of CC3^+^ cells based on immunostaining is easier and there is less variation in different field so we took the average per mouse. The use of two quantification methods gave use more confidence in the data.

The data has now been put in order and described. We added description for Figure 1C and added more description for Figure 1D.

2) "E2B treatment alone without the antagonist (E2B+DMSO) lead to an expected 1.57-fold increase (p=0.0082) in mammary neutrophils as compared to the Ctrl+DMSO". Should "1.57-fold" be "2.57-fold" or something other? It is not the case based on the data in Figure 3Ci.

Again thank you for pointing out the mistake. The number have been corrected to 2.57 and 2.26, respectively.

3) In Figure 4B, upon neutrophil depletion, Cebpb and Cebpd were already increased, which could limit their further enhancement when treated with E2B. As for Adig and Egr2, it seemed that they also apparently increased. In Figure 4D, the data had the similar problem to those in Figure 4B. No description about Figure 4E and 4F was found in the content. Overall, these data put it in question that estrogen-induced adipocyte repopulation is associated with the induction of adipogenic and tissue remodeling genes through neutrophils.

We agree with the reviewer’s interpretation of *Cebpb* and *Cebpd* data. Figure 4 has been now rearranged and described accordingly.

4) "This suggests that the up-regulation of Ctsb expression by E2B is a direct event independent of STAT3 activation". These data in Figure 5B could not demonstrate that Ctsb expression is the direct event of E2B. In Figure 5D, why the lysosomal pellet fractions showed no lysosomal proteins, such as catheptins. In Figure 5A, at least the protein level of TNF-α should be measured, because it was very important for the functions of E2B, based on the data in Figure 6.

Our interpretation is based on the following. First, STAT3 activation has been shown to induce cathepsin expression during mammary involution [35]. Although we did not detect an increase in p-STAT3 with estrogen treatment, we observed a significant increase in *Ctsb* expression in response to estrogen. This implies that *Ctsb* is up-regulated by mechanism other than Stat3. Second, *Ctsb* is a known estrogen target gene that is also induced in nulliparous mice (Figure 7). Therefore, we rationalize that estrogen induced *Ctsb* expression was independent of STAT3 activation.

Cathepsin was not detected in Figure 5D in the pellet (lysosomal) fraction because during mammary involution, there is increased lysosomal leakiness and cathepsins are all leaked to the cytosol [35]. Therefore, cathepsin was only detected in the supernatant fraction but not in the pellet fraction after sub-cellular fractionation.

We tried ELISA to detect TNF-α in mammary cell lysate. But the signals are too low to allow a reliable evaluation. On the other hand, the upregulation of TNF-α gene expression by estrogen has been consistent in different experiment settings including an experiment in our previous report [2].

5) In Figure 6, TNF induces p-STAT3 while Figure 5A shows E2 induces TNF expression (mRNA), but no p-STAT3 was increased in Figure 5B. The increased mRNA does not mean the increased protein. Please measure the TNFα proteins in Figure 5A (See Issue#4). The MCF-7/Casp3 model seems not to well support the conclusion. The data in Figure 6 are about typical apoptosis not the lysosomes-associated cell death involved in the functions of estrogen as revealed in this study.

The data in Figure 6 was meant to demonstrate that under an inflammatory condition, estrogen (which is more commonly known as a mitogen) can stimulate cell death. But we did not expect it to be via LM-PCD because MCF7-Caspase3(+) cells are different in many ways from the mouse involuting mammary tissue.

**Reference**

https://www.guidetopharmacology.org/GRAC/ObjectDisplayForward?objectId=69

1. Shen, H., D. Kreisel, and D.R. Goldstein, Processes of sterile inflammation. J Immunol, 2013. **191**(6): p. 2857-63.2. Chung, H.H., et al., Estrogen reprograms the activity of neutrophils to foster protumoral microenvironment during mammary involution. Scientific Reports, 2017. **7**.3. O'Brien, J., et al., Macrophages are crucial for epithelial cell death and adipocyte repopulation during mammary gland involution. Development, 2012. **139**(2): p. 269-75.4. Pepe, G., et al., Self-renewal and phenotypic conversion are the main physiological responses of macrophages to the endogenous estrogen surge. Sci Rep, 2017. **7**: p. 44270.5. Daley, J.M., et al., Use of Ly6G-specific monoclonal antibody to deplete neutrophils in mice. J Leukoc Biol, 2008. **83**(1): p. 64-70.6. Sue, R.D., et al., CXCR2 is critical to hyperoxia-induced lung injury. J Immunol, 2004. **172**(6): p. 3860-8.7. Bertini, R., et al., Noncompetitive allosteric inhibitors of the inflammatory chemokine receptors CXCR1 and CXCR2: prevention of reperfusion injury. Proc Natl Acad Sci U S A, 2004. **101**(32): p. 11791-6.8. Belperio, J.A., et al., CXCR2/CXCR2 ligand biology during lung transplant ischemia-reperfusion injury. J Immunol, 2005. **175**(10): p. 6931-9.9. Cugini, D., et al., Inhibition of the chemokine receptor CXCR2 prevents kidney graft function deterioration due to ischemia/reperfusion. Kidney Int, 2005. **67**(5): p. 1753-61.10. ; Available from: .11. Stauffer, S.R., et al., Pyrazole ligands: structure-affinity/activity relationships and estrogen receptor-α-selective agonists. J Med Chem, 2000. **43**(26): p. 4934-47.12. Meyers, M.J., et al., Estrogen receptor-β potency-selective ligands: structure-activity relationship studies of diarylpropionitriles and their acetylene and polar analogues. J Med Chem, 2001. **44**(24): p. 4230-51.13. Foghsgaard, L., et al., Cathepsin B acts as a dominant execution protease in tumor cell apoptosis induced by tumor necrosis factor. Journal of Cell Biology, 2001. **153**(5): p. 999-1009.14. Luzio, J.P., P.R. Pryor, and N.A. Bright, Lysosomes: fusion and function. Nat Rev Mol Cell Biol, 2007. **8**(8): p. 622-32.15. Kreuzaler, P.A., et al., Stat3 controls lysosomal-mediated cell death in vivo. Nature cell biology, 2011. **13**(3): p. 303-9.16. Di Donato, N., et al., Mutations in CRADD Result in Reduced Caspase-2-Mediated Neuronal Apoptosis and Cause Megalencephaly with a Rare Lissencephaly Variant. Am J Hum Genet, 2016. **99**(5): p. 1117-1129.17. Vanden Berghe, T., et al., Necroptosis, necrosis and secondary necrosis converge on similar cellular disintegration features. Cell Death Differ, 2010. **17**(6): p. 922-30.18. Kono, H. and K.L. Rock, How dying cells alert the immune system to danger. Nat Rev Immunol, 2008. **8**(4): p. 279-89.19. Orozco, S., et al., RIPK1 both positively and negatively regulates RIPK3 oligomerization and necroptosis. Cell Death Differ, 2014. **21**(10): p. 1511-21.20. Nakazawa, D., et al., Activated platelets induce MLKL-driven neutrophil necroptosis and release of neutrophil extracellular traps in venous thrombosis. Cell Death Discov, 2018. **4**: p. 6.21. Rock, K.L. and H. Kono, The inflammatory response to cell death. Annu Rev Pathol, 2008. **3**: p. 99-126.22. Zhao, L., et al., Pharmacological activation of estrogen receptor β augments innate immunity to suppress cancer metastasis. Proc Natl Acad Sci U S A, 2018. **115**(16): p. E3673-E3681.23. Bengtsson, A.A., et al., Pharmacokinetics, tolerability, and preliminary efficacy of paquinimod (ABR-215757), a new quinoline-3-carboxamide derivative: studies in lupus-prone mice and a multicenter, randomized, double-blind, placebo-controlled, repeat-dose, dose-ranging study in patients with systemic lupus erythematosus. Arthritis Rheum, 2012. **64**(5): p. 1579-88.24. Bjork, P., et al., Identification of human S100A9 as a novel target for treatment of autoimmune disease via binding to quinoline-3-carboxamides. PLoS Biol, 2009. **7**(4): p. e97.25. Deronic, A., et al., The quinoline-3-carboxamide paquinimod (ABR-215757) reduces leukocyte recruitment during sterile inflammation: leukocyte- and context-specific effects. Int Immunopharmacol, 2014. **18**(2): p. 290-7.26. Kraakman, M.J., et al., Neutrophil-derived S100 calcium-binding proteins A8/A9 promote reticulated thrombocytosis and atherogenesis in diabetes. J Clin Invest, 2017. **127**(6): p. 2133-2147.27. Wache, C., et al., Myeloid-related protein 14 promotes inflammation and injury in meningitis. J Infect Dis, 2015. **212**(2): p. 247-57.28. Werneburg, N.W., et al., Tumor necrosis factor-α-associated lysosomal permeabilization is cathepsin B dependent. American journal of physiology. Gastrointestinal and liver physiology, 2002. **283**(4): p. G947-56.29. Hennigar, S.R. and S.L. Kelleher, TNFalpha Post-Translationally Targets ZnT2 to Accumulate Zinc in Lysosomes. J Cell Physiol, 2015. **230**(10): p. 2345-50.30. Hennigar, S.R., et al., ZnT2 is a critical mediator of lysosomal-mediated cell death during early mammary gland involution. Sci Rep, 2015. **5**: p. 8033.31. Droga-Mazovec, G., et al., Cysteine cathepsins trigger caspase-dependent cell death through cleavage of bid and antiapoptotic BCl^-^2 homologues. J Biol Chem, 2008. **283**(27): p. 19140-50.32. Stoka, V., et al., Lysosomal protease pathways to apoptosis. Cleavage of bid, not pro-caspases, is the most likely route. J Biol Chem, 2001. **276**(5): p. 3149-57.33. Janicke, R.U., et al., Caspase-3 is required for DNA fragmentation and morphological changes associated with apoptosis. J Biol Chem, 1998. **273**(16): p. 9357-60.34. Yang, X.H., et al., Reconstitution of caspase 3 sensitizes MCF-7 breast cancer cells to doxorubicin- and etoposide-induced apoptosis. Cancer Res, 2001. **61**(1): p. 348-54.35. Kreuzaler, P.A., et al., Stat3 controls lysosomal-mediated cell death in vivo. Nat Cell Biol, 2011. **13**(3): p. 303-9.